# The Zugspitze radiative closure experiment for quantifying water vapor absorption over the terrestrial and solar infrared.
# Part III: Quantification of the mid- and near-infrared water vapor continuum in the 2500 to 7800 cm$^{-1}$ spectral range under atmospheric conditions

Andreas Reichert and Ralf Sussmann

Karlsruhe Institute of Technology, IMK-IFU, Garmisch-Partenkirchen, Germany

*Correspondence to:* A. Reichert (andreas.reichert2@kit.edu)

**Abstract.** We present a first quantification of the near-infrared (NIR) water vapor continuum absorption from an atmospheric radiative closure experiment carried out at Mt. Zugspitze (47.42° N, 10.98° E, 2964 m a.s.l.). Continuum quantification is achieved via radiative closure using radiometrically calibrated solar FTIR absorption spectra covering the 2500 to 7800 cm$^{-1}$ spectral range. The dry atmospheric conditions at the Zugspitze site (IWV 1.4 to 3.3 mm) enable continuum quantification even within water vapor absorption bands, while upper limits for continuum absorption can be provided in the centers of window regions. Throughout 75 % of the 2500 to 7800 cm$^{-1}$ spectral range, the Zugspitze results are agree within our estimated uncertainty with the widely used MT_CKD 2.5.2-model (Mlawer et al., 2012). In the wings of water vapor absorption bands, our measurements indicate about 2-5 times stronger continuum absorption than MT_CKD, namely in the 2800 to 3000 cm$^{-1}$ and 4100 to 4200 cm$^{-1}$ spectral ranges. The measurements are consistent with the laboratory measurements of Mondelain et al. (2015), which rely on cavity ring-down spectroscopy (CDRS), and the calorimetric-interferometric measurements of Bicknell et al. (2006). Compared to the recent FTIR laboratory studies of Ptashnik et al. (2012, 2013), our measurements are consistent within the estimated errors throughout most of the spectral range. However, in the wings of water vapor absorption bands our measurements indicate typically 2 – 3 times weaker continuum absorption under atmospheric conditions, namely in the 3200 to 3400 cm$^{-1}$, 4050 to 4200 cm$^{-1}$, and 6950 to 7050 cm$^{-1}$ spectral regions.

## 1 Introduction

Atmospheric water vapor is the most important contributor to the absorption of incoming solar radiation in the near infrared (NIR, 4000—14000 cm$^{-1}$) (Kiehl and Trenberth, 1997). Water vapor absorption comprises both the effect of spectral line absorption and the broadband so-called continuum absorption (e.g. Shine et al., 2012). Depending on the atmospheric state and the choice of continuum model, up to 6% of the clear-sky water vapor absorption can be attributed to the continuum (Paynter and Ramaswamy, 2011). Consequently, quantitative knowledge of this contribution is a prerequisite for realistic atmospheric radiative transfer calculations employed e.g. in climate models (Paynter and Ramaswamy, 2014; Rädel et al., 2015; Turner et al., 2012).

However, the NIR atmospheric water vapor continuum currently still lacks sufficient experimental constraints. Recently, a number of laboratory studies based on different experimental techniques investigated this open question. Several efforts were made to quantify continuum absorption, including the contributions of both the self and foreign-broadened continuum. Several studies made use of cell measurements with grating spectrometers (e.g. Burch 1982; 1985; Burch and Alt 1984) or FTIR (Fourier Transform Infrared) spectrometers (Baranov et al., 2008; Baranov and Lafferty, 2011; Paynter et al., 2009; Ptashnik et al. 2011; 2012; 2013; and 2015). Furthermore, a number of spectral regions were covered by cavity ring-down spectroscopy (CRDS) measurements (Cormier et al. 2002, 2005; Mondelain et al., 2013, 2014, 2015), by the related technique of optical feedback cavity enhanced spectroscopy (OF-CEAS, Ventrillard et al., 2015) and by

calorimetric-interferometric measurements (Fulghum and Tilleman, 1991; Bicknell et al., 2006). However, no consensus has been reached among these studies. As noted e.g. by Mondelain et al. (2014) and Ptashnik et al. (2013), the individual results feature differences far beyond the respective uncertainty estimates whose attribution to causative processes remains tentative. A further challenge for laboratory studies is that they are typically carried out at higher temperatures than those encountered in the atmosphere in order to detect the weak continuum absorption in the limited optical path length of the cells. Note that CDRS and related techniques in principle enable measurements at atmospheric temperature (see e.g. Cormier et al., 2005) but such measurements are not yet available for many spectral regions). To date, the temperature dependence of the self-continuum has been investigated by measurements in a number of spectral regions (e.g. Cormier et al., 2005; Mondelain et al., 2014; Ptashnik et al., 2011; Ventrillard et al., 2015). However, the remaining uncertainty of the self-continuum temperature dependence (see e.g. Paynter and Ramaswamy, 2011) and the lack of measurements of the foreign continuum temperature dependence cause considerable uncertainties in the application of the laboratory results on atmospheric radiative transfer calculations.

The continuum has been investigated for atmospheric conditions using measurements of atmospheric emitted infrared radiance for other spectral regions (e.g. Tobin et al., 1999; Rowe and Walden, 2009). However, atmospheric measurements are available only for a fraction of the spectral region covered by this study (Newman et al. 2011; 2400 to 3200 cm-1), while for the remaining interval from 3200 to 7800 $cm^{-1}$, no atmospheric measurements have been reported. A validation of continuum absorption strength under atmospheric conditions is therefore highly desirable to address these shortcomings. To this aim, we conducted a radiative closure experiment with the objective of quantifying the NIR water vapor continuum absorption from atmospheric measurements. The study is carried out at the high-altitude Zugspitze site and relies on the solar FTIR measurements implemented at this site (Sussmann and Schäfer, 1997) in the framework of the Network of the Detection of Atmospheric Composition Change (NDACC, www.ndacc.org). While such atmospheric closure studies enable to avoid some limitations of laboratory measurements as outlined above, they are also subject to a number of major challenges: absorption in the NIR due to aerosols can become comparable to the magnitude of the water vapor continuum absorption of interest (Ptashnik et al., 2015) and requires an accurate separation of continuum and aerosol contribution. Furthermore, the characterization of the atmospheric state (e.g. IWV, water vapor profile, temperature profile, and further trace gas column amounts) is more challenging and typically less accurate than the characterization of experimental conditions in a laboratory study.

This paper is part of a three-paper series about different aspects of the Zugspitze radiative closure experiment. The first paper, hereafter referred to as Part I (Sussmann et al., 2016, same issue), describes the instrumental setup, evaluates the sensitivity of the closure experiment in the far infrared (FIR, 2—667 $cm^{-1}$), the mid-infrared (MIR, 667—4000 $cm^{-1}$), and the NIR, and provides results on the FIR water vapor continuum. A novel radiometric calibration method for solar FTIR spectra in the NIR is presented in a second paper, referred to as Part II (Reichert et al., 2016, same issue). Part III (this paper) contains the NIR continuum quantification method and results. Continuum quantification in the NIR is achieved comparing calibrated radiance spectra, obtained with the method presented in Part II, to radiative transfer model calculations. The results derived from our data set are presented and compared to results from laboratory studies as well as the widely used MT_CKD 2.5.2 continuum model (Mlawer et al., 2012).

This paper is structured as follows: Section 2 contains an overview of the instrumental setup used in the closure experiment. Section 3 outlines the method for water vapor continuum quantification. In Sect. 4, the results obtained with this method are presented and compared to previous studies. Finally, Sect. 5 contains a summary and conclusions.

## 2 Setup of the closure experiment

The closure experiment relies on a quantitative comparison of measurements of spectral radiance with synthetic spectra calculated using the line-by-line radiative transfer model (LBLRTM, Clough et al., 2005). Spectral line parameters were set according to the aer_v3.2 line list provided alongside the LBLRTM model. Water vapor continuum absorption is then quantified via the spectral residuals, i.e. the difference between simulated and measured spectra. We adopt the definition of the water vapor continuum given in Turner et al. (2010), i.e. water vapor continuum is defined as all absorption by water vapor exceeding a Voight line shape within $\pm\ 25$ cm$^{-1}$ of each line center minus the value of the Voight line shape at $\pm\ 25$ cm$^{-1}$ ("plinth").

The instruments used in the Zugspitze radiative closure experiment and the related uncertainties are described in detail in Part I. In summary, spectral radiances in the NIR are measured using a solar FTIR spectrometer setup at the Zugspitze (47.42° N, 10.98° E, 2964 m a.s.l.) summit observatory (Sussmann and Schäfer, 1997). Radiative calibration of measured spectra is achieved via a novel calibration method presented in Part II, which relies on a combination of the Langley method and measurements of a medium-temperature blackbody source.

The atmospheric state at the time of the radiance measurements is required as an input to the LBLRTM radiative transfer calculations. To enable accurate quantification of the water vapor continuum from spectral residuals, the atmospheric state has to be constrained precisely using a number of additional measurements listed in the following.

Vertically integrated water vapor (IWV) constitutes the key input parameter and is derived directly from the solar FTIR spectra (e.g. Sussmann et al. 2009; Schneider et al., 2012). Temperature and pressure profiles are taken from four-times-daily National Center for Environmental Prediction (NCEP) resimulation data. NCEP resimulation data is also used to constrain the shape of the water vapor profile. Column-averaged mixing ratios of $CO_2$, $CH_4$, and $N_2O$ are measured using the nearby Garmisch TCCON (Total Carbon Column Observing Network) solar FTIR instrument (Sussmann and Rettinger, 2014). $O_3$ columns were constrained by combined Brewer/Dobson measurements made at the nearby Hohenpeissenberg observatory (Köhler, 1995).

Aerosol optical depth (AOD) has to be constrained precisely in order to enable continuum quantification in the wings of the strong NIR water vapor bands and within window regions. A great advantage of the Zugspitze site is that AOD is typically very low, i.e. the AOD for the Zugspitze dataset is about a factor of 10 lower than at typical lowland mid-latitude sites. The AOD levels encountered in our closure data set (for data set description and selection criteria see Sect. 3.3) are in the range 0.0005 - 0.00075 at 2500 cm$^{-1}$ and in the range 0.0024 - 0.0032 at 7800 cm$^{-1}$ at airmass 1. AOD was measured using the SSARA-Z (Sun-Sky Automatic Radiometer - Zugspitze) sun photometer (Toledano et al., 2009) developed by the Meteorological Institute of the University of Munich and set up at Schneefernerhaus (2675 m a.s.l., 680 m horizontal distance to the Zugspitze solar FTIR). The instrument includes 13 spectral channels from 340 to 1640 nm. Only information from 5 channels whose central wavelengths are in the spectral region between 439.6 and 781.1 nm was used in the analysis. The exact filter wavelengths and full width at half maximum (FWHM) values of these channels are listed in Table 1. The reason for the channel selection is that in the ultra-violet (UV) to visible range, water vapor continuum absorption can be considered negligible compared to AOD, whereas for the NIR channels continuum absorption will lead to biased AOD results. The channels below 440 nm were excluded since the high influence of Rayleigh scattering in the UV leads to increased AOD uncertainties.

The data analysis of the SSARA-Z measurements was implemented similar to the approach outlined by Toledano et al. (2009). In detail, we used standard Langley calibration for cloud-free periods. Rayleigh scattering was accounted for using the formula given by Bodhaine et al. (1999). In the analysis, a Gaussian shape was assumed for the filter transmissivity curves. The influence of absorption by $O_3$ was subtracted as outlined in Guyemard et al. (1995). NIR AOD was then deduced by assuming AOD wavelength dependence according to the Ångstrom relation:

$$\tau(\lambda) = b \cdot \lambda^{-\alpha}, \tag{1}$$

where $\tau$ designates AOD. The Ångstrom exponent $\alpha$ and scaling $b$ are determined by a fit to the UV/visible AOD measurements. More sophisticated descriptions of the AOD wavelength dependence such as the relation given by Molineaux et al. (1998) may be used instead of Eq. (1). However, the number of sun photometer wavelength channels included in our analysis is not sufficient to place tight constraints on the higher number of parameters used in such models. Furthermore, the very low AOD at Zugspitze leads to high relative errors in the AOD determined from sun photometer measurements, which removes the benefits of more advanced models compared to Eq. (1).

The AOD uncertainty comprises several contributions: First of all, the AOD determined from the sun photometer measurements is affected by uncertainty in the radiance measurements. This uncertainty contribution was set according to the 2-$\sigma$ radiance measurement noise. The calibration uncertainty ensues from the uncertainty of the Langley fit. Additional uncertainty arises from the Rayleigh scattering correction, where central wavelength and FWHM errors of optical filters and atmospheric pressure errors contribute. The treatment of $O_3$ absorption is also prone to additional errors, due to filter parameter and $O_3$ column errors. In addition to these contributions, further uncertainty is induced by the fit to Eq. (1) that enables constraining the NIR AOD from the UV/visible measurements. The overall AOD uncertainty that ensues from these contributions for our data set at air mass 1 is <0.0015 at 2500 cm$^{-1}$ and <0.0025 at 7800 cm$^{-1}$.

## 3 NIR continuum determination

### 3.1. Method overview

The aim of this study is to constrain the NIR water vapor continuum absorption under atmospheric conditions. We make use of the radiative closure experiment setup at the Zugspitze observatory that is described in detail in Part I. Generally, radiative closure experiments comprise a quantitative comparison of spectral radiance measurements to synthetic spectra. The strategy for water vapor continuum quantification employed in this study relies on radiometrically calibrated solar FTIR spectra in the 2500 to 7800 cm$^{-1}$-range. An alternative method that has been proposed by Mlawer et al. (2014) and relies on the Langley method is presented in Appendix C.

Spectra were recorded with the solar FTIR instrument described in Sect. 2 and Part I, using no optical filter, a spectral resolution of 0.02 cm$^{-1}$ (resolution is defined as 0.9/optical path difference), and averaging over 4-8 scans which leads to a 75-150 s repeat cycle per spectrum. The measured spectra are radiometrically calibrated by means of the calibration method outlined in Part II. Briefly, the calibration approach relies on Langley calibration in suitable spectral windows with little atmospheric absorption. In addition to the Langley technique that enables highly accurate calibration in selected windows, the shape of the calibration curve between the windows is constrained using spectral radiance measurements of a high-temperature blackbody source. The calibration uncertainty achieved with this novel method is 1-1.7 % (2 $\sigma$) throughout the spectral range considered. Synthetic radiance spectra are generated using the LBLRTM radiative transfer model. Figure 1 shows the mean measured and synthetic radiance spectra for the closure data set that will be presented in Sect. 3.3. The atmospheric state used as an input to the calculations was set based on the measurements described in Sect. 2. Given the calibrated spectral radiance measurements and the synthetic spectra, radiance residuals $\Delta\boldsymbol{I}$ can then be calculated for a set of spectra selected according to the criteria that will be presented in Sect. 3.3

$$\Delta\boldsymbol{I} = \boldsymbol{I}_{\text{FTIR}} - \boldsymbol{I}_{\text{LBLRTM, no continuum}} \cdot e^{-\text{AOD}}, \tag{2}$$

where $\boldsymbol{I}_{\text{FTIR}}$ designates the radiometrically calibrated solar FTIR spectra, $\boldsymbol{I}_{\text{LBLRTM, no continuum}}$ the synthetic LBLRTM spectra not including continuum absorption and AOD the aerosol optical depth. Continuum optical depth $\boldsymbol{\tau}_{\text{cont}}$ is calculated from the spectral residuals as follows:

$$\boldsymbol{\tau}_{\text{cont}} = -\ln\left(\frac{\Delta\boldsymbol{I}}{\boldsymbol{I}_{\text{LBLRTM, no continuum}} \cdot e^{-\text{AOD}}} + 1\right). \tag{3}$$

After calculation of the continuum optical depth (OD), absorption coefficients were derived from these results. The continuum OD $\tau_{cont}$ is linked to continuum absorption coefficient $k_{cont}$ as follows

$$\tau_{cont} = m \cdot \int_{h_{obs}}^{\infty} k_{cont}(T, n_{wv}, n_{air}) \cdot n_{wv} dh, \tag{4}$$

where $m$ designates the relative air mass, $h_{obs}$ the altitude of the observing instrument, $n_{wv}$ the water vapor number density, and $n_{air}$ the dry air number density.

$k_{cont}$ can be further decomposed in self- and foreign continuum contributions according to

$$k_{cont} = c_s \cdot \frac{\rho_{H_2O}}{\rho_0} + c_f \cdot \frac{\rho_{air}}{\rho_0} , \tag{5}$$

where $c_s$ and $c_f$ designate the self- and foreign continuum coefficients and $\rho_{H_2O}$, $\rho_{air}$, and $\rho_0$ are the densities of water vapor, dry air and a reference density, respectively. In detail, $\rho_0 = P_0/(k_b T_0)$, where $P_0 = 1013$ mbar, $k_b$ is the Boltzmann constant, and $T_0 = 296$ K. In addition to their different dependence on water vapor density according to Eq. 5, self- and foreign broadened continua are characterized by their distinct temperature dependence: while the self continuum shows strong negative temperature dependence, the foreign continuum is assumed to have no or only weak temperature dependence.

The separation of $k_{cont}$ in self- and foreign continuum contributions from atmospheric measurements is challenging. In principle, an assignment to self- and foreign continuum is possible using a large set of measurements covering a wide range of atmospheric conditions, i.e. IWV and temperature. However, the available data does not permit such an assignment given the sensitivity of our setup as discussed in Sect. 4. Therefore, in the following, we characterize continuum strength using the mean continuum absorption coefficient $\overline{k}_{cont}$, defined as follows:

$$\overline{k}_{cont} = \frac{\int_{h_{obs}}^{\infty} k_{cont}(T, n_{wv}, n_{air}) \cdot n_{wv} dh}{\int_{h_{obs}}^{\infty} n_{wv} dh,} = \frac{\tau_{cont}}{m \cdot IWV} \tag{6}$$

Low-uncertainty constraints on $\overline{k}_{cont}$ can only be placed in a number of spectral windows. The selection of such suitable windows is outlined in Sect. 3.4. The continuum results for each spectrum were computed as the median of $\overline{k}_{cont}$ in all selected spectral windows within 10 cm$^{-1}$-wide bins. Finally, an error-weighted mean of $\overline{k}_{cont}$ was calculated from the set of 52 spectra selected according to the criteria listed in Sect. 3.3. The uncertainty estimate of the continuum results is presented in Sect. 3.2.

### 3.2. Uncertainty estimate

An interpretation of the residual OD and assignment to causative absorption processes requires a comprehensive uncertainty budget of the closure experiment. The uncertainty estimate of our experimental setup is described in detail in Part I except for contributions only relevant for the NIR closure measurements. The total residual uncertainty and its various contributions are also shown in Part I, Fig. 5. A description of the NIR-specific contributions and a brief outline of the remaining sources of uncertainty are given below. All uncertainty values are quoted on 2-$\sigma$ confidence level.

i) Absorption line parameter uncertainties of water vapor and other absorbing species. These uncertainties were set to the mean value of the uncertainty range specified by the error codes provided in the line parameter file (aer_v3.2) provided alongside the LBLRTM model. Line parameter uncertainties are the dominant contribution to the error budget within absorption bands.

ii) A further significant contribution to the error budget results from the IWV measurement uncertainty. The IWV precision was set to 0.8 %, the bias to 1.1 % according Schneider et al. (2012). The uncertainty resulting from NCEP water vapor profile shape errors was estimated using a comparison of NCEP profiles to radiosonde data (see Part I for details).

iii) The OD uncertainty resulting from NCEP temperature profile errors was deduced from a temperature error covariance matrix estimate for the NCEP resimulation profiles. The error covariance matrix estimate was constructed from the comparison of coincident NCEP profiles to a radiosonde campaign conducted at the site (see Part I for details).

iv) Column uncertainties for further trace gases (e.g. $CO_2$, $CH_4$, $N_2O$ and O3) are also included in the uncertainty estimate. The respective column accuracies are listed in Part I (Tab. 2 therein).

v) The AOD uncertainty is of crucial importance for the OD uncertainty budget in the window regions. As outlined in Sect. 2, the AOD uncertainty at air mass 1 is < 0.0025 for the closure data set throughout the 2500 to 7800 cm$^{-1}$-range.

The uncertainty contributions i) to v) listed above are linked to the accuracy of the atmospheric state input for LBLRTM calculations. Aside from that, an additional group of error contributions stems from the solar FTIR spectral radiance measurements:

vi) The radiance uncertainty due to the radiometric calibration is about 1 – 1.7% and is described in detail in Part II.

vii) A further uncertainty contribution results from the solar FTIR measurement noise. It is determined directly from solar FTIR spectra and is among the few uncertainty contributions in the closure experiment of strictly statistical character. It is therefore largely reduced by taking mean results from a larger set of spectra.

viii) Ice layer formation on the liquid nitrogen cooled InSb detector can occur in case of leaks in the detector's vacuum enclosure. Ice formation leads to additional absorption in certain spectral regions, most notably in the 3000 to 3400 cm$^{-1}$-range. The uncertainty contribution by varying ice absorption was estimated using lamp spectra routinely recorded with the solar FTIR. Variations in ice absorption during the time period covered by the experiment can be detected as a change of the ratio of measured signal outside and inside the ice absorption band. The maximum variation of this ratio detected in the lamp spectra (1.6 %) was taken as an estimate of the error due to ice absorption.

ix) Only a fraction of the solar tracker mirrors is covered by the instrument's field of view (FOV). Due to non-ideal alignment of optical elements, the exact location of the area observed by the instrument on the mirror changes depending on the azimuth and elevation of the instrument's line of sight. The reflectivity of the tracker mirrors features spatial inhomogeneity due to dirt and aging effects. In combination with the moving area covered by the FOV, this results in a variation in measured radiance which leads to spurious variations in the measured OD. An estimate of this uncertainty contribution can be gained using an outgoing laser beam aligned with the instrument's optical axis that enables constraining the mirror area covered by the FOV depending on the instrument's azimuth and elevation. A detailed description of this analysis is given in Part II, Sect. 4.1.

x) The uncertainty due to inaccuracies in the ESS was estimated from repeating the continuum retrieval using the ESS versions by Kurucz (2005) and Menang et al. (2013), which differ by about 5 % (see Sect. 4). The corresponding uncertainty contribution corresponds to 11.1% of the remaining continuum uncertainty budget on average (see also Fig. 9 of the companion publication Part I).

### 3.3. Spectra selection

We analyzed spectra recorded under cloud-free conditions in the December 2013 – February 2014 period. Due to inaccuracies in the air mass calculation at high solar zenith angle, air mass was required to be below $m = 9.0$.

In Sect. 3.2, we outlined a source of radiance error in the solar FTIR measurements due to the pointing variation on the tracker mirrors and give an estimate of this contribution. For spectra included in the closure data set, this uncertainty contribution was required to be negligible compared to other sources of uncertainty, in detail the selection threshold was set to a maximum radiance error of 0.1%. These selection criteria lead to a final dataset of 52 selected solar FTIR spectra covering an IWV range from 1.4 to 3.3 mm for which the continuum results are presented in Sect. 4. The mean atmospheric state of the closure data set is listed in Appendix A.

### 3.4. Micro-window selection

To select suitable windows for continuum quantification, a number of selection criteria were applied to the spectra. Several criteria make use of upper or lower envelopes to the spectra, which were constructed as follows: The upper/lower envelope is defined as the linear interpolation between the highest/lowest values encountered within each 10 cm$^{-1}$-wide wavenumber bin. In detail, the following filtering criteria were applied to the spectra:

i) To avoid spectral regions affected by line absorption, only the spectral points with the lowest OD compared to the surrounding spectral region were used. In detail, only points for which the OD exceeds the lower envelope by less than the 2-$\sigma$ OD uncertainty were used.

ii) Regions around solar lines were excluded. This was implemented as an exclusion of all points for which the extra-atmospheric solar radiance according to the ESS of Kurucz (2005) is more than 0.5 % below the upper envelope. Note that recent studies indicate that many solar lines are missing in this ESS (see Menang et al., 2013). However, solar lines omitted in the ESS of Kurucz (2005) are discarded from further analysis by applying the selection criterion (i). As outlined in Sect. 4, a repetition of the continuum analysis using the ESS of Menang et al. (2013), which includes many additional solar lines, only leads to very minor changes in the continuum results, thereby indicating that the solar line removal scheme according to criteria (i) and (ii) is appropriate.

iii) Only regions with low OD uncertainty are included. Therefore, we select points less than 10 % above the lower envelope to the uncertainty.

iv) In order to avoid biases of the retrieved continuum due to measurement noise, only regions with a signal-to-rms-noise ratio $s/n > 5$ were included.

The selection thresholds cited above were adjusted in order to provide sufficiently dense coverage with selected points while maintaining optimum selection quality. Different experimental setups may therefore require different selection threshold values. The final continuum OD results were computed as the median value of all selected spectral points within 10 cm$^{-1}$-wide bins.

### 4 Results

Figure 1 shows the mean continuum absorption coefficient $\overline{k}_{cont}$ determined from the Zugspitze dataset in comparison to the MT_CKD 2.5.2 model predictions and several recent laboratory studies. The figure includes laboratory measurements carried out at or below room temperature which provided constraints on both the self and foreign continuum using the same or a very similar experimental setup. The mean atmospheric state of the closure dataset is listed in Appendix A, while a table with our results for $\overline{k}_{cont}$ and the relative scaling of our results vs. the MT_CKD 2.5.2 predictions and associated uncertainties are available as supplementary material (Supplement A). The results shown in Fig. 2 are in very good agreement with the $\overline{k}_{cont}$-values derived using the Langley method as outlined in Appendix C. The assignment of the residual OD to water vapor continuum absorption was made based on two arguments: As outlined in Part I, great care was taken to construct a comprehensive uncertainty budget including thorough estimates of all relevant error contributions to the closure experiment. Therefore, contributions to the residual OD from other processes than water vapor continuum absorption far beyond the indicated error bars seem unlikely. Furthermore, the IWV dependence of the measured residual OD is consistent with that expected from water vapor continuum absorption. As $\overline{k}_{cont}$ includes contributions due to both foreign- and self continuum, it is expected to scale as the sum of a constant and a linear term with respect to water vapor density and therefore also with respect to IWV. The closure dataset covers an IWV range of 1.4 mm < IWV < 3.5 mm, which enables investigation of the IWV dependence of $\overline{k}_{cont}$. Due to the narrow range of atmospheric temperatures covered in the data set, temperature dependence of the self continuum can be neglected in this analysis. A fraction of 98.6 % of all measured continuum

absorption coefficients in the Zugspitze data set are consistent with a combination of constant and linear scaling with respect to IWV, i.e. with being caused by a combination of foreign- and self water vapor continuum. However, 94.2 % of the data are also consistent with a purely constant scaling, i.e. with being solely due to foreign continuum absorption. This is due to the fact that at the atmospheric conditions covered by the data, in all spectral regions where continuum absorption is detectable beyond the experiment's sensitivity, the foreign continuum constitutes by far the dominant contribution, assuming that the partitioning in self- and foreign continuum given by the MT_CKD model is approximately correct. Note that this assumption has to be considered tentative since for both self- and foreign continuum the results of recent laboratory studies deviate from the MT_CKD model especially in window regions (e.g. Ptashnik et al, 2012, 2013; Mondelain et al., 2015). Examples of the measured $\overline{k}_{cont}$ and the best fit constant and linear scaling for wavenumber bins within water vapor bands, in the wings of bands and in window regions are shown in Fig. A3.

This analysis shows that the contribution of the self continuum is not unambiguously detectable due to the limited sensitivity of our experiment. We therefore provide values of the mean continuum absorption coefficient $\overline{k}_{cont}$ as defined by Eq. (6), including contributions from both self- and foreign continuum instead of the more commonly used continuum coefficients $c_s$ and $c_f$. If values for $c_f$ are required, further assumptions on the self continuum have to be made before subtracting this contribution. As an example, Supplement B to this manuscript contains a list of $c_f$-values for all spectral bins where $c_f$ exceeds the uncertainty estimate. The results were calculated from our measurements assuming the self continuum to be consistent with the MT_CKD model. Recent laboratory measurements (e.g. Ptashnik et al., 2013) suggest that this assumption may not be appropriate. However, alternative sources of the self continuum neither constitute a more robust estimate, given the inconsistencies between different laboratory results, the uncertainty of the self continuum temperature dependence and the fact that the foreign continuum is likely to be the dominant contribution to the overall continuum absorption for the dry atmospheric conditions of our study and the spectral windows covered by the measurements (see Fig. 1). A 50 % uncertainty was assumed for the self continuum as suggested by Paynter et al. (2011) and is included in the uncertainty of $c_f$ in addition to the uncertainty budget presented in Sect. 3.2.

As outlined in the companion paper Part II, recent studies on the NIR ESS have yielded results that feature differences of up to 5-10% (see e.g. Menang et al, 2013; Bolsee et al, 2014; Thuillier et al., 2014, 2015; Weber et al. 2015). Furthermore, the number of solar lines differs significantly e.g. between the ESS versions of Kurucz (2005) and Menang et al., (2013). To investigate the influence of inaccuracies in the ESS on the continuum results, the continuum retrieval was repeated using the ESS determined by Menang et al. (2013) instead of the ESS by Kurucz (2005) that was used to generate the results presented in Fig. 2. This is a good test to assess the sensitivity of the results to ESS uncertainty since these ESS versions differ by about 5 %, while recent ESS results generally feature differences of up to ±5 % compared to the ESS of Kurucz (2005). Note that the Menang et al. (2013) ESS only covers the spectral region > 4000 cm$^{-1}$. The comparison is therefore restricted to 4233 cm$^{-1}$ < ν < 7800 cm$^{-1}$, which corresponds to the first Langley point covered by the Menang et al. (2013) ESS and the maximum wavenumber value covered by our analysis. For this region, the median of the absolute value of the difference between the Menang et al. (2013) and Kurucz (2005) continuum results corresponds to 11% of the continuum uncertainty estimate. Therefore, ESS uncertainty does not constitute a major accuracy limitation of our analysis, which his due to the fact that the same ESS is used for both synthetic spectra calculation and the radiometric calibration presented in the companion paper Part II.. The ESS-related continuum uncertainty was estimated from the difference of the Menang et al. (2013) and Kurucz (2005) results and included in the uncertainty budget as described in Sect. 3.2 (see also Fig. 9 of the companion paper Part I). For the spectral region ν < 4233 cm$^{-1}$, where no direct comparison is available, the ESS-induced continuum uncertainty was assumed to correspond to 11% of the remaining overall uncertainty as suggested by the median value in the spectral range ν > 4233 cm$^{-1}$.

The prediction of the MT_CKD 2.5.2 model is shown alongside our results for $\overline{k}_{cont}$ in Fig. 1. The MT_CKD 2.5.2-values of $\overline{k}_{cont}$ were computed in an analogous way as the values derived from our dataset, i.e. $\overline{k}_{cont}$ was calculated according to Eq. (6)

for the set of atmospheric states encountered in the data set. The results shown in Fig. 1 represent the mean of the MT_CKD predictions for the set of selected measurements. Overall, there is good agreement of our results with the MT_CKD values. Consistency within a 2-$\sigma$ range is observed for 75 % of the spectral range covered by our measurements. The most apparent discrepancy between MT_CKD and our results occurs in the 2800 to 3000 cm$^{-1}$-range, where our results are about a factor of 5 higher than the MT_CKD predictions. However, care has to be taken in the interpretation of this discrepancy since the 2800 to 3000 cm$^{-1}$ spectral range coincides with a methane absorption band. Therefore, the accuracy of the continuum result in this range depends on whether the HITRAN error estimate for methane line parameters is correct and whether line coupling effects where treated in a sufficiently realistic way in the LBLRTM model. Further significant discrepancies ensue in the 4100 to 4200 cm$^{-1}$ wavenumber region. The higher measurement results from the Zugspitze data indicate that the MT_CKD-model underestimates the continuum absorption in the wings of the 4000 to 5000 cm$^{-1}$ window region. In the centers of water vapor absorption bands (i.e. ~5200-5400 cm$^{-1}$ and ~7100-7300 cm$^{-1}$), our results are significantly lower than the MT_CKD-predictions for a number of spectral points. However, the continuum results in these regions are highly sensitive to accurate input and uncertainty estimates for IWV and water vapor line parameters. Therefore, the slight differences found in the band centers do not provide robust evidence for necessary adjustments of the MT_CKD model.

Figure 1 also includes a comparison of our results to several current laboratory studies using different experimental approaches for continuum quantification. For the comparison, $\overline{k}_{cont}$-values were calculated for the same set of atmospheric states as our results using the continuum coefficients given in the respective studies. For the Mondelain et al. (2015) and Bicknell et al. (2006) results, we used the MT_CKD temperature dependence. Since the results of Bicknell et al. (2006) do not allow a dissociation of self from foreign continuum is not possible, we assumed the self-to-foreign ration suggested by the MT_CKD model to calculate the corresponding value of $\overline{k}_{cont}$. The self continuum temperature dependence proposed by Rädel et al. (2015), which , which was deduced from the measurements of Ptashnik et al. (2011), was used for all laboratory studies. Note, however, that the importance of the continuum temperature dependence is limited (5 to 20 %, see below) for our dataset. This is due to the fact that no temperature dependence is assumed for the foreign continuum, which is by far dominant for most spectral regions given the dry atmospheric conditions encountered in our data set. A fraction of the spectral range covered by this study, namely 2500-3200 cm$^{-1}$, was also included in the airborne measurements by Newman et al. (2011). Newman et al. (2011) conclude that the increase of the self continuum in MT_CKD 2.5 compared to MT_CKD 2.4 lead to reduced spectral residuals, while no firm conclusion can be drawn in the 2500 -3200 cm$^{-1}$ –range on whether MT_CKD 2.5 or the results of Ptashnik et al. (2011) represent a more appropriate quantitative description of the water vapor self continuum. These findings are in agreement to the results of this study, given that both are not consistent with continuum absorption being weaker than indicated by MT_CKD 2.5. Our results show very good agreement with the CDRS-based measurements of Mondelain et al. (2015). Due to the dominant role of the foreign continuum in the 4100 to 4200 cm$^{-1}$ spectral range, this agreement mainly corresponds to a comparison of the foreign continuum results of Mondelain et al. (2015) and our measurements. Therefore, our results are consistent with the finding of Mondelain et al. (2015) and Ptashnik et al. (2012) that the foreign continuum is underestimated by the MT_CKD model in this spectral region. For the spectral range examined by Bicknell et al. (2006) with calorimetric-interferometric measurements, only the upper limit of the continuum absorption is constrained by our data, which is consistent with all laboratory studies cited here. The comparison of our results to the BPS_MTCKD 2.0 continuum proposed by Paynter and Ramaswamy (2014) is mostly equivalent to the comparison to MT_CKD. This is due to the fact that the BPS_MTCD 2.0 foreign continuum, which constitutes the dominant contributor for the dry atmospheric conditions encountered in our data set, was mostly adopted from MT_CKD. Exceptions include the spectral regions from 2500 to 3000 cm$^{-1}$, 5200 to 5600 cm$^{-1}$, and 6800 to 7000 cm$^{-1}$, where our results show better consistency with the MT_CKD 2.5.2 model. The FTIR-based results of Ptashnik et al. (2012, 2013) in combination with the temperature dependence proposed by Rädel et al. (2015) lead to higher absorption coefficients than our data in several spectral regions. Significant inconsistencies beyond the uncertainty range occur mostly in the wings of water vapor absorption bands, e.g. in the 3200 to 3400 cm$^{-1}$, and 4000 to 4200 cm$^{-1}$ ranges as visible in Fig. 1. In these ranges the absorption coefficients provided

by the FTIR laboratory measurements are typically a factor of 2-3 higher compared to our data. Further FTIR laboratory measurements were carried out by Baranov and Lafferty (2011) at temperatures of 311 to 363 K on the self continuum and by Baranov (2011) at 326 to 363 K on the foreign continuum at $\nu < 3500$ cm$^{-1}$. The results of these studies generally agree well within the estimated errors with the findings of Ptashnik et al. (2012, 2013). As noted e.g. by Ptashnik et al. (2015), weak lines not included in the line list used for the synthetic spectra calculation may bias the retrieved continuum results. This effect is largely reduced in our analysis due to the spectral selection criteria applied, namely the selection of low-OD windows as outlined in Sect. 3.4, criterion i). An issue not accounted for in our analysis is the uncertainty of the continuum temperature dependence, since an uncertainty estimate is provided neither for the MT_CKD nor the Rädel et al. (2015) relations. However, under the atmospheric conditions covered by our data set and assuming the MT_CKD self-to-foreign ratio, the self continuum contributes only 10 to 30 % to the total continuum absorption at the spectral points for which we detect significant continuum absorption. While no temperature dependence is assumed for the dominant foreign contribution, the temperature dependence of the self continuum changes the mean continuum absorption coefficient by 5 to 20 % within the spectral range considered here and assuming the Rädel et al. (2015) relation. Therefore, it seems unlikely that the differences between the results of Ptashnik et al. (2012) and (2013) and our data are solely due to inaccuracies in the continuum temperature dependence. Note, however that the assumption that the foreign continuum has no significant temperature dependence, which was used in the data analysis, has not been robustly confirmed by measurements under atmospheric conditions yet. Due to the dominant role of the foreign continuum in the wings of water vapor absorption bands, inaccuracies in the foreign continuum temperature dependence would have a significant influence on the conversion of the findings of Ptashnik et al. (2012, 2013) to atmospheric temperatures.

## 5 Summary and conclusions

We present a quantification of the water vapor continuum absorption in the NIR spectral range (2500 to 7800 cm$^{-1}$) from an atmospheric radiative closure experiment. To our knowledge, prior to this study no precise constraints on the continuum absorption under atmospheric conditions were available for most of this spectral range. The mean continuum absorption coefficient was determined from a set of 52 solar FTIR spectra. The method to achieve continuum quantification relies on the use of radiometrically calibrated spectra obtained by the method presented in Part II. Continuum constraints are presented in the wings and some spectral windows in the centers of water vapor absorption bands. Due to the low IWV encountered throughout our measurement period, only the upper boundary of the continuum can be constrained in the centers of atmospheric windows.

The results show good consistency with the widely used MT_CKD 2.5.2 model, although they indicate a need for increasing the absorption strength compared to the model in some spectral regions such as the wings of water vapor absorption bands. Our results were compared to a number of recent laboratory studies using different experimental techniques. A first group of studies relies on FTIR cell measurements. Our data generally indicate lower continuum absorption than implicated by the studies of Ptashnik et al. (2012, 2013) in combination with the self continuum temperature dependence given in Rädel at al. (2015). However, significant deviations from these studies only occur in the wings of water vapor absorption bands. There are also several regions where our results are in good agreement with the findings of Ptashnik et al. (2012, 2013), most notably around 3000 cm$^{-1}$. Further experimental techniques used for continuum quantification in laboratory experiments include CRDS. A comparison to the CDRS results of Mondelain et al. (2015) in the spectral region around 4250 cm$^{-1}$ shows very good agreement to our findings. Bicknell et al. (2006) quantified continuum absorption using a calorimetric-spectrometric technique. While our results agree to the findings of Bicknell et al. (2006), their measurements cover spectral regions where only an upper limit for the continuum absorption can be deduced from our data.

An assignment of the detected continuum absorption to self- and foreign continuum requires improvements of the experimental sensitivity or a data set covering a broader range of IWV values. The same is true for a detection of the

continuum beyond the uncertainty limit in window regions which requires improved sensitivity or a data set covering higher IWV values. Aside from these limitations, our results provide a valuable foundation for an improved quantification of the NIR water vapor continuum under atmospheric conditions. Most notably, our analysis provides a tool for atmospheric validation of the predictions of current laboratory studies and the MT_CKD continuum model in the NIR spectral range. This is of crucial importance since the results of recent studies carried out using different experimental techniques show inconsistent results and to date no experimental validation under atmospheric conditions was available.

## Appendix A: Mean atmospheric state of the closure data set

## Appendix B: Supplementary figures

## Appendix C: Continuum quantification using the Langley method

Langley measurements (see e.g. Liou, 2002 and Part II) are part of the radiometric the calibration method employed in this study which is presented in the companion paper Part II. These Langley measurements can be used to directly quantify the water vapor continuum as outlined by Mlawer et al. (2014), thereby offering a validation of the results obtained with the method presented in Sect. 3. For continuum quantification according to this strategy, spectrally resolved atmospheric optical depth is determined using the Langley method for all wavenumber values according to the analysis scheme presented in Part II. The atmospheric optical depth without water vapor continuum absorption is then calculated using the LBLRTM model using the atmospheric state input set as outlined in Sect. 3 and the companion paper Part I. Residual optical depth between Langley result and model calculation in spectral windows selected according to Sect. 3.4 is interpreted as water vapor continuum absorption.

The OD uncertainty budget required for the window selection in the case of the Langley method is largely similar to the uncertainty estimate described in Sect. 3.2. Differences include the absence of the calibration uncertainty, which is replaced by the OD uncertainty of the Langley fit, including the uncertainty contribution due to air mass inaccuracies. Furthermore, the error due to changing ice absorption on the detector is not included in the error budget, since, by construction, the Langley method is only sensitive to atmospheric absorption. The mean continuum absorption coefficient is then calculated from the residual OD as described in Sect. 3.1.

Figure C1 shows the mean continuum absorption coefficients derived from the Langley measurements made on 12 December 2013 (black data points). The results are compared to the MT_CKD 2.5.2 model (blue line) and the results obtained with the calibrated method according to Sect. 3. (orange data points), which were calculated for the same set of 16 spectra used in the Langley method. Throughout 98.0 % of spectral range for which results from both methods are available, the absorption coefficients are consistent within the 2 σ error estimate. While in window regions both methods are equally suitable for continuum quantification, large errors due to air mass uncertainties make the use of the calibrated method presented in Sect. 3 preferable in the vicinity of water vapor absorption bands.

*Acknowledgements.* We thank for the constructive and helpful referee and short comments, which led to significant improvements of this manuscript. We furthermore thank H. P. Schmid (KIT/IMK-IFU) for his continual interest in this work. Funding by the Bavarian State Ministry of the Environment and Consumer Protection (contracts TLK01U-49581 and VAO-II TP I/01) and Deutsche Bundesstiftung Umwelt is gratefully acknowledged. It is our pleasure to thank E. Mlawer (AER) for suggesting the exploitation of our Zugspitze solar FTIR measurements for NIR continuum quantification, which is the subject of this paper. The authors are indebted to D.D. Turner (NOAA) for helpful conversations during the definition

phase of the Zugspitze radiance closure project. We furthermore thank Ulf Köhler (Meteorologisches Observatorium Hohenpeissenberg, DWD) for providing ozone column measurements, Matthias Wiegner (LMU München) for access to sun photometer measurement data and Petra Hausmann (KIT/IMK-IFU) for providing IWV retrievals. We are grateful for support by the Deutsche Forschungsgemeinschaft and Open Access Publishing Fund of the Karlsruhe Institute of Technology.

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

# Tables

**Tab. 1:** Central wavelength and FWHM of the sun photometer (SSARA-Z) filters used for AOD analysis.

| $\lambda$ [nm] | FWHM [nm] |
|---|---|
| 439.6 | 9.7 |
| 498.7 | 12.3 |
| 531.9 | 11.2 |
| 672.5 | 10.9 |
| 781.1 | 9.7 |

**Tab. A1:** Mean atmospheric state of the closure data set: pressure-, temperature- and water vapor density profiles. The data set was selected from Zugspitze solar FTIR spectra measured from Dec 2013 – Feb 2014 and contains 52 spectra. Spectra selection criteria are listed in Sect. 3.3.

| altitude [km a.s.l] | $P$ [mbar] | $T$ [K] | $\rho_{\text{water vapor}}$ [g/m$^3$] |
|---|---|---|---|
| 2.964 | 714.074 | 270.570 | 1.270 |
| 2.975 | 713.085 | 270.522 | 1.266 |
| 2.987 | 712.008 | 270.469 | 1.263 |
| 3.009 | 710.036 | 270.372 | 1.256 |
| 3.032 | 707.982 | 270.271 | 1.249 |
| 3.066 | 704.946 | 270.121 | 1.239 |
| 3.099 | 702.000 | 269.974 | 1.229 |
| 3.147 | 697.763 | 269.671 | 1.201 |
| 3.262 | 687.664 | 268.899 | 1.127 |
| 3.497 | 667.380 | 267.256 | 0.966 |
| 3.600 | 658.649 | 266.536 | 0.898 |
| 3.700 | 650.259 | 265.838 | 0.833 |
| 3.800 | 641.950 | 265.139 | 0.769 |
| 3.900 | 633.727 | 264.440 | 0.707 |
| 4.000 | 625.592 | 263.741 | 0.645 |
| 4.100 | 617.538 | 263.042 | 0.585 |
| 4.200 | 609.570 | 262.342 | 0.526 |
| 4.300 | 601.680 | 261.644 | 0.468 |
| 4.400 | 593.871 | 260.919 | 0.446 |
| 4.500 | 586.147 | 260.186 | 0.436 |
| 4.600 | 578.503 | 259.455 | 0.425 |
| 4.700 | 570.935 | 258.723 | 0.415 |
| 4.800 | 563.441 | 257.991 | 0.405 |
| 4.900 | 556.027 | 257.260 | 0.395 |
| 5.000 | 548.693 | 256.528 | 0.385 |
| 5.500 | 513.149 | 252.871 | 0.338 |
| 6.000 | 479.451 | 249.366 | 0.261 |
| 6.500 | 447.548 | 245.966 | 0.169 |
| 7.000 | 417.379 | 242.567 | 0.087 |
| 8.000 | 361.924 | 235.748 | 0.027 |
| 9.000 | 312.542 | 228.923 | $9.69 \cdot 10^{-3}$ |
| 10.00 | 268.744 | 222.463 | $2.96 \cdot 10^{-3}$ |
| 15.00 | 123.703 | 213.438 | $4.17 \cdot 10^{-4}$ |
| 20.00 | 54.6496 | 209.890 | $2.35 \cdot 10^{-4}$ |
| 30.00 | 10.775 | 212.607 | $6.27 \cdot 10^{-5}$ |
| 40.00 | 2.488 | 248.853 | $1.41 \cdot 10^{-5}$ |
| 60.00 | 0.179 | 239.829 | $9.17 \cdot 10^{-7}$ |
| 100.0 | $2.77 \cdot 10^{-4}$ | 213.601 | $1.13 \cdot 10^{-10}$ |
| 120.0 | $2.38 \cdot 10^{-5}$ | 378.719 | $8.74 \cdot 10^{-13}$ |

**Tab. A2:** Mean atmospheric state of the closure data set: trace gas column amounts. The data set was selected from Zugspitze solar FTIR spectra measured from Dec 2013 – Feb 2014 and contains 52 spectra. Spectra selection criteria are listed in Sect. 3.3.

| | |
|---|---|
| IWV | 2.26 mm |
| $XCO_2$ | 395.3 ppm |
| $XCH_4$ | 1781 ppb |
| $XN_2O$ | 311.8 ppb |
| $O_3$ column | 279.9 DU |

**Figures**

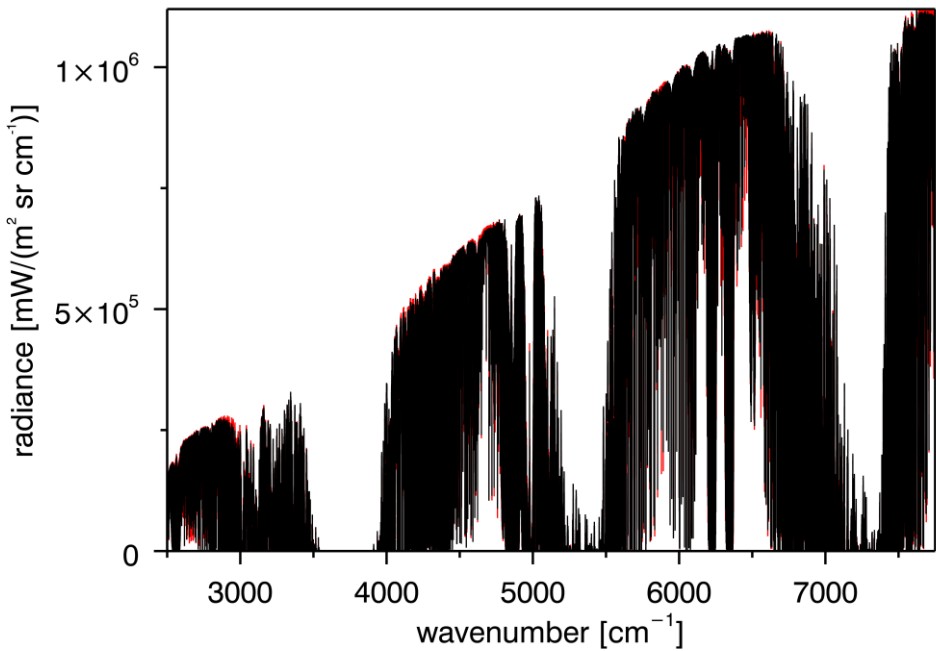

**Fig. 1**: Mean measured (black) and synthetic (red) radiance spectra for the closure data set selected according to the criteria presented in Sect. 3.3.

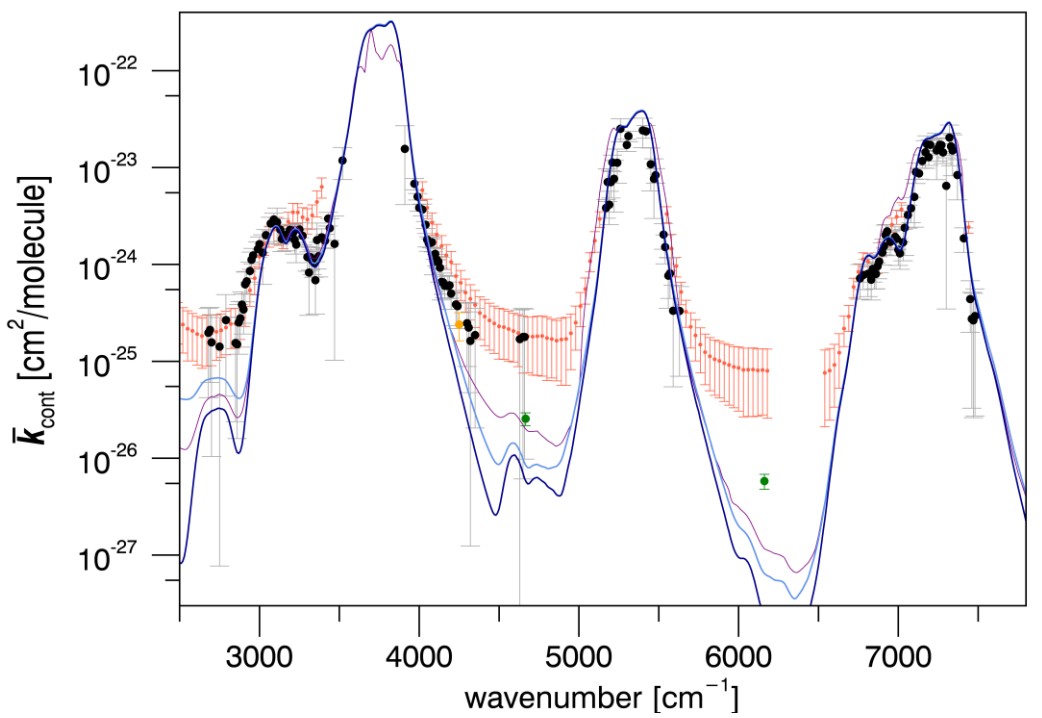

**Fig. 2:** Mean continuum absorption coefficient $\overline{k}_{\mathrm{cont}}$ determined in the Zugspitze closure experiment and corresponding 2-$\sigma$ uncertainties (black). The figure only includes data points for which the measured continuum exceeds the estimated uncertainty. A representation of the full set of measurement results is shown in Fig. B1. Results are compared to the MT_CKD 2.5.2 model (self and foreign continuum: light blue, foreign continuum: dark blue), the BPS-MTCKD 2.0 model (purple line) and the following laboratory studies carried out at room temperature or below: the CRDS measurements of Mondelain et al. 2015 (orange), the calorimetric-interferometric measurements of Bicknell et al. (2006) (green) and the FTIR measurements of Ptashnik et al. (2012, 2013) (red).

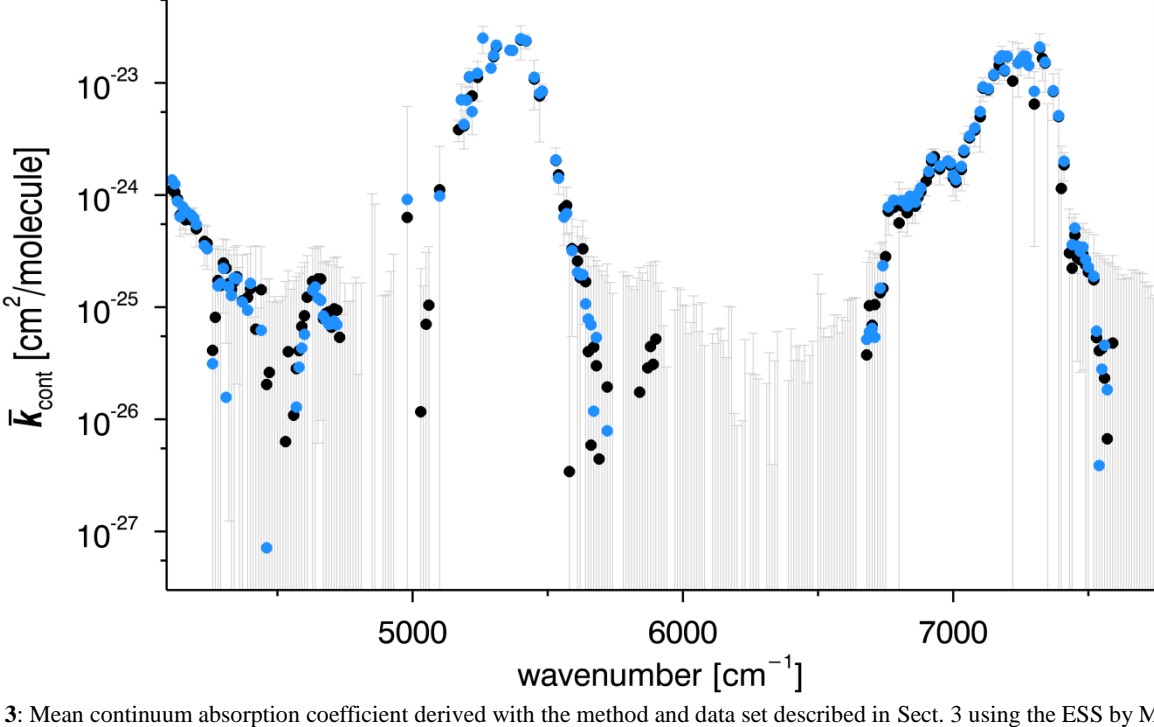

**Fig. 3**: Mean continuum absorption coefficient derived with the method and data set described in Sect. 3 using the ESS by Menang et al. (2013) (blue data points) and by Kurucz (2005) (black data points). The different ESS sources differ by about 5 % and many solar lines not present in Kurucz (2005) were included in Menang et al. (2013).

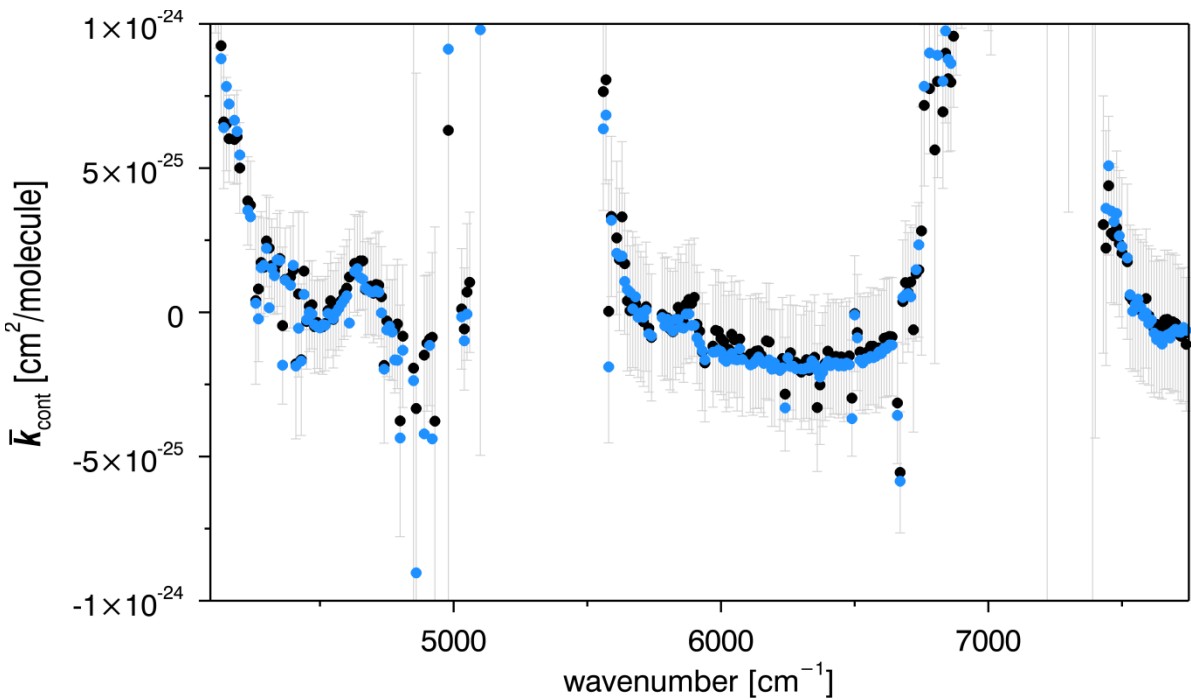

**Fig. 4**: Same as Fig. 3 but using a linear scale to show the results in the window regions.

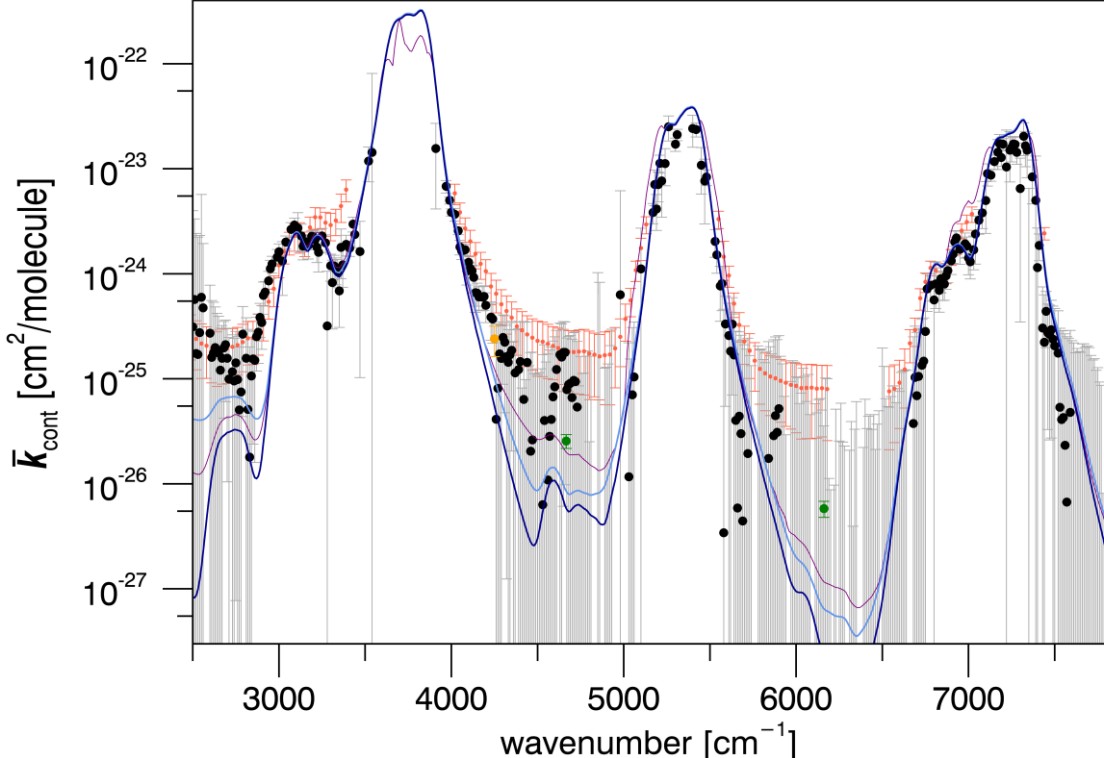

**Fig. B1:** Mean continuum absorption coefficient $\bar{k}_{cont}$ determined in the Zugspitze closure experiment and corresponding 2-$\sigma$ uncertainties (black). Results are compared to the MT_CKD 2.5.2 model (self and foreign continuum light blue, foreign continuum dark blue), the BPS-MTCKD 2.0 model (purple line) and the following laboratory studies carried out at room temperature or below: the CRDS measurements of Mondelain et al. 2015 (orange), the calorimetric-interferometric measurements of Bicknell et al. (2006) (green) and the FTIR measurements of Ptashnik et al. (2012, 2013) (red).

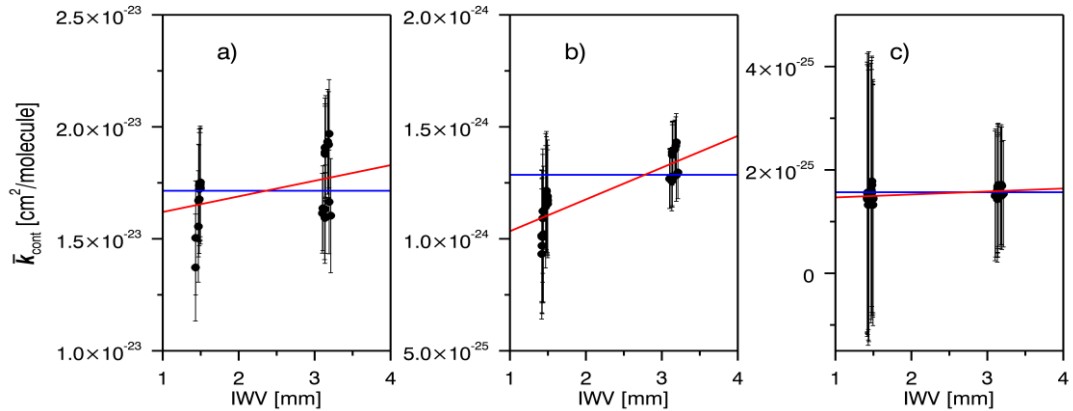

**Fig. B2:** Examples of the scaling of the mean continuum absorption coefficient $\bar{k}_{cont}$ (black data points) with respect to IWV a) within a water vapor absorption band (4100 cm$^{-1}$), b) in the wings of an absorption band (4700 cm$^{-1}$), and c) in a window region (2700 cm$^{-1}$). The blue line corresponds to the best fit constant scaling, the red line to the best fit linear scaling.

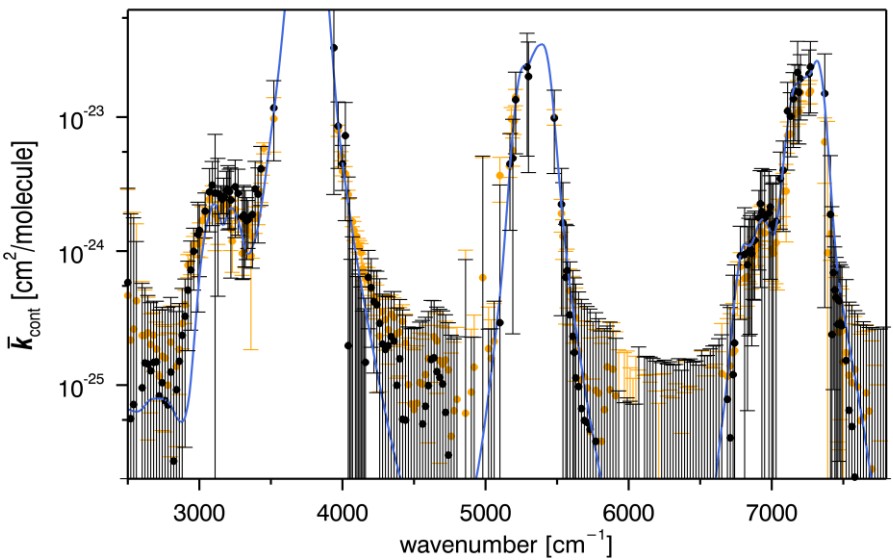

**Fig. C1.:** Mean continuum absorption coefficient $\overline{k}_{cont}$ determined from 12 December 2013 spectra using the Langley method and corresponding $2\,\sigma$ uncertainties (black). Results are compared to the calibrated method for the same spectral dataset (orange) and the MT_CKD 2.5.2 model (blue).

