# Peer review of "The Zugspitze radiative closure experiment for quantifying water vapor absorption over the terrestrial and solar infrared. Part III: Quantification of the mid- and near-infrared water vapor continuum in the 2500 to 7800 cm-1 spectral range under atmospheric conditions"

_Atmospheric Chemistry and Physics, 2016_

## Referee Comment (RC1) · Anonymous Referee #1 · 23 May 2016

Journal: ACP
Title: The Zugspitze radiative closure experiment for quantifying water vapor absorption over the terrestrial and solar infrared.  Part III: Quantification of the near-infrared water vapor continuum under atmospheric conditions
Author(s): A. Reichert and R. Sussmann
MS No.: acp-2016-323
MS Type: Research article
Special Issue: Twenty-five years of operations of the Network for the Detection of Atmospheric Composition Change (NDACC) (AMT/ACP/ESSD inter-journal SI)

**General comments:**

In this work, the authors measured the absorption due to water vapor contribution from solar FTIR absorption spectra recorded at Mt Zugspitze. This was done using the radiative closure method.

The paper is well-written and the level of details given is generally sufficient taking into account the fact that two companion papers describing the setup and the calibration procedure would be published in the same Special Issue. The site of Mt Zugspitze has the advantage of reduced aerosol optical density compared to typical lowland mid-latitude sites minimizing the impact of aerosols on the continuum determination. Great care is taken to consider all the possible sources of uncertainty which is important for the method used here. This leads to large uncertainties bars in the windows where the continuum is weak. In Supplementary material the authors give mean water vapor continuum absorption coefficients with negative values which have no physical meaning. Moreover data are given with large uncertainty in the center of the windows so that they are in agreement with all the literature data within the error bars bringing no additional information for those spectral regions. They also weaken the other data obtained in the bands and at the edges of the windows with lower uncertainty. To me it will have more sense to remove large uncertainty data in the center of the 2.1 µm, 1.6 µm windows before publication.

As mentioned in the companion paper Part 1: *Very dry atmospheric conditions are a pre-requisite for closure studies of this kind due to the otherwise saturated spectral regions*. This has for consequences that in these conditions of dry air, the foreign-continuum represents most of the continuum absorption (more than 70% if the MT_CKD self to foreign ratio is assumed according to the authors). This can be viewed as a kind of "limitation" of the method but it is also important as there is a real lack of observational constraints for the foreign-continuum.

To conclude, this paper presents state-of the art atmospheric measurements of the continuum which bring interesting information in the near infrared bands and at the edges of the windows mostly for the foreign-continuum for which experimental constraint are clearly missing. For these reasons this paper deserves to be published in ACP after large uncertainty data in the center of the 2.1 µm and 1.6 µm windows are removed.

**Specific comments:**

**P1, L32:** Burch (1982) and Burch and Alt (1984) used a grating spectrometer for their experiments and not a FTIR spectrometer.

The following references are missing for continuum measured:
**By CRDS**:
-   Mondelain D. et al., J. Quant. Spectrosc. Radiat. Transfer, 130, 381 (2013).
-   Cormier, J. G. et al., J. Chem. Phys., 122, 114309, (2005)
-   Cormier J. G., et al., J. Chem. Phys. 116,1030 (2002)
**By OF-CEAS**:
-   Ventrillard et al. J. Chem. Phys. 143, 134304 (2015)
**By calorimetric-interferometry** :
-   Fulghum, S. F., and M. M. Tilleman, J. Opt. Soc. Am. B Opt. Phys., 8, 2401(1991)

**P1, L38:** The sentence is too general. Cavity enhanced techniques like CRDS and OF-CEAS as well as CI are able to measure continuum absorption in the windows at room temperature (see references above) and even at lower temperature (see Cormier 2005).

**P2, L1-3:** The sentences "To date…non-straightforward" are too definitive. Temperature dependences of the self-continuum cross-section have been investigated in different spectral windows (see for example Cormier 2005, Mondelain 2014, Ptashnik 2011, Ventrillard 2015…). In the 2.1 μm window, for example, the temperature dependence measured at high temperatures by the CAVIAR consortium is similar to that measured at lower temperature by Ventrillard et al. in different part of the window. For sure there is a *real* lack of observational constraints for the foreign-continuum.

**P2, L4-5:** To me it is much more difficult to characterize the continuum from atmospheric spectra than from laboratory spectra recorded in well-known conditions of temperature with sufficiently sensitive and stable techniques. In atmospheric conditions additional uncertainties occur due to water vapor profile, temperature profile, aerosols uncertainties and it is more difficult to separate self and foreign continuum.

**P2, L23:** Which version of the HITRAN database is used for the line-by-line calculations? Is the line profile used is a Voigt profile truncated at +/- 25 cm$^{-1}$ from the line center with the plinth subtracted? If yes the authors have to mention it.

**P7, L32:** How exactly calculations are done in the case of Bicknell et al. (2006) as these measurements did not allow dissociating the self from foreign continuum.

**P7, L8:** Assuming that the partitioning in self and foreign continuum given by the MT_CKD model is a quite strong hypothesis as the very few laboratory studies of the foreign-continuum (Ptashnik PTRSA 2012 and Mondelain PCCP 2015) seem to show that the foreign continuum cross-sections are largely underestimated by MT_CKD in the windows.

**P7, L38:** The very good agreement with CRDS-based measurements of Mondelain et al. (2015) is essentially a good agreement with the foreign-value measured in this paper due to the dominating contribution of the foreign-continuum in the atmospheric conditions encountered at Zugspitze. This is an important point as in the Mondelain et al study the measured foreign-cross section is 4.5 times larger than the one given in MT_CKD 2.5. The fact that in the 4100 to 4200 cm$^{-1}$ spectral range the MT_CKD model underestimate the continuum goes in the same direction as well as the Ptashnik 2012 paper. This point should be underlined by the authors.

**Fig 1:** Some literature data at room temperature are not plotted on this figure (Ventrillard 2015, Mondelain 2014). It will be good to incorporate them in the figure.

**Technical corrections:**
**P2, L25**: *closure experiment **und** the related uncertainties*: und->and.

---

## Referee Comment (RC2) · P. Rowe (Referee) · 6 Jun 2016

Overall comments: This article presents water vapor continuum coefficients in the spectral range 2500 to 7800 cm-1 from solar FTIR absorption spectra. The authors find that their measurements agree well with the widely-used MT-CKD continuum for most of the range, but that their results are 5 times stronger from 2800 to 3000 and 4100 to 4200 cm-1. Compared to recent laboratory measurements, they find 2–5 times weaker continuum absorption for 3200 to 3400 cm$-1$, 4050 to 4200 cm$-1$, and 6950

to 7050 cm−1 spectral regions.

These measurements are (to the authors' knowledge) the first in this spectral range for atmospheric conditions (but note a small overlap with previous work mentioned below). The results are thus relevant and interesting. The error analysis is thorough and the article is well written. I recommend publication with minor changes, as noted below:

Main comments:

1) Literature review. I do not think it is necessary to exhaustively reference laboratory work performed over a different spectral range. But all lab work that overlaps the spectral region should be referenced. (Thus I think some, but not all, of the references suggested by the other reviewer need to be included). In addition, studies in atmospheric conditions with similar instruments should be referenced (although they are mainly in different spectral regions):

Page 2, line 5. Please include the following references with a sentence such as, "The continuum has been investigated for atmospheric conditions using measurements of atmospheric emitted infrared radiance for other spectral regions (Tobin et al., 1999; Rowe and Walden, 2009) and for part of the region of interest for this study (Newman et al. 2011; 2400 to 3200 cm-1), but not for the spectral region 3200 to 7800 cm-1."

- Tobin et al.: Downwelling spectral radiance observations at the SHEBA ice station: Water vapor continuum measurements from 17 to 26 micron, J. Geophys. Res., 104, 2081-2092, 1999.

- Rowe, P.M. and Walden, V.P.: Improved measurements of the foreign-broadened continuum of water vapor in the 6.3 micron band at -30 C, Appl. Opt. 48, 1358-1365, 2009.

- Newman et al. 2012: Airborne and satellite remote sensing of the mid-infrared water vapour continuum, Phil. Trans. R. Soc. A 370, 2611-2636.

The results of Newman et al. should also be discussed (e.g. on Page 7, line 38). Also

on Page 8, line 20, change "in this spectral range" to "for most of this spectral range."

2) Fig. 1. If you also include kcont for the self and foreign-broadened parts of the MT-CKD continuum separately in this plot, it will show the relative importance of each for your results. If any of the lower bounds on your error bars are significantly different from zero, you might make those error bars black instead of gray so they stand out more. In the caption, "(black)" needs to be changed to "(black; gray error bars are shown for points for which only the upper threshold can be determined to within the uncertainty)." If you have measurements that are not shown on this plot, you should create a second panel below on a linear y-scale where they can be seen. In the caption, state something like: "x points that fall outside the plot region are not shown in the upper panel (log scale) but are evident in the lower panel (linear scale)"

3) (Optional) The paper would likely receive more citations if the following changes were made.

- Title: Stating the spectral range explicitly in the title will help readers more quickly determine if the paper is of interest to them, especially given the large spectral range. (You could omit "under atmospheric conditions" because the title already includes "radiative closure experiment.")

- Cf. If you estimate the foreign-broadened part of the continuum (Cf), it can be compared to previous work, and incorporated (e.g. in figures) in future publications by other authors. I suggest estimating the self-broadened continuum (Cs) based on what you think is most accurate (MT-CKD or other; give rationale) and the atmospheric water and temperature structure, removing its effects from kcont, and calculating Cf. Increase your error bars correspondingly (uncertainty estimate for self-broadened continuum x 0.1 to 0.3). Discuss how you calculated Cf briefly in the text. Add a figure showing the subset of results for which the error bars are small enough to be useful.

Minor comments

**ACPD**

- Give the wavenumber range the first time you mention each region (near-infrared, etc)

- Page 4, Line 8: Please show examples of measured and synthetic radiance spectra. (You can alternatively reference your other paper here, but I think it would be nice to have it here as well).

- Page 4, line 20 (approx.). You might mention here that cs is strongly temperature dependent but that cf is thought to be only weakly temperature dependent.

Technical and grammar corrections

- Page 1, Line 26, remove the word "exact"

- Page 1, Line 31, change "both ... continuum" to "continuum absorption, including the contributions of both the self and foreign-broadened continuum"

- Page 2, Line 10, change "thereafter" to "hereafter"

- Page 3, Line 4, add the word "for" before "data"

- Page 3, line 8, change "disposes" to "consists of" or "includes"

- Page 3, line 9, rephrase "centered at ... nm." Perhaps give the range of the channels or the bounds of each.

- Page 3, line 24, Do you mean errors in the AOD measurements from the sun photometer (rather than errors in the sun photometer measurements)? If so, add "AOD determined from" before "sun photometer measurements."

- Page 3, lines 26-27, "The following ... measurements." The sentence is awkward, rephrase.

- Page 4, line 11, change "the criteria presented" to "criteria that will be presented"

- Page 6, line 6, change "requested" to "required"

- Page 7, line 21, change "where treated in sufficiently" to "were treated in a sufficiently"

- Page 8, line 18, change "presented" to "present"

- Page 9, line 12, change "we thank for support by the" to "we are grateful for support by the"

-Table A1. Convert into two tables, putting further parameters in a separate table.

---

## Short Comment (SC1) · 14 Jun 2016

**Comment on "The Zugspitze Radiative Closure Experiment .... Part III" by A.Reichert and R. Sussman (doi: 10.5194/acp-2016-323)**

Comment by Keith P Shine1\*, Jonathan Elsey1 and Igor Ptashnik2

1Department of Meteorology, University of Reading, Reading RG6 6BB, UK

2Atmospheric Spectroscopy Division, V.E. Zuev Institute of Atmospheric Optics Tomsk, 634021, Russia

\*email: k.p.shine@reading.ac.uk

Submitted: 15 June 2016

Reichert and Sussman (2016) present an important attempt to characterise the water vapour continuum in the near-infrared in atmospheric conditions. Given that relatively few such measurements exist, such work is very welcome.

We have a number of comments on the paper. The major one relates to our comment on Part II of this paper, where the authors calibrate their measurements to an assumed extraterrestrial solar spectrum (ESS); as we note in that comment, there are significant uncertainties in the ESS. This uncertainty has important consequences for the derivation of the continuum, especially in the window regions, which are not taken into account here.

It is our view that this uncertainty renders the continuum derivations here unreliable in window regions; the fact that many of the derived continuum values in the windows are negative and therefore unphysical (as shown in the data in their Supplement but not in the figure in the paper) adds support to the opinion given by Reviewer 1 (10.5194/acp-2016-323-RC1) that the derived continuum values deep in the window are so uncertain that they should not be presented.

**Major comments**

1. Equations (2) and (3) derive the continuum optical depth from the difference between the observed downward radiance at the surface and the modelled radiance ignoring the continuum. To do this reliably requires that the ESS is well constrained. This is not currently the case, as we explain in our comment in Part II (see e.g. Thuillier et al. (2015) and Weber (2015)). Various derivations from satellite and other observations differ by 5-10%.

The authors' method is essentially to write a radiance residual (their Equation (2)) between observations and model so that

$$\Delta I = S_{actual} \exp(-(\tau_g + \tau_{cont} + \tau_{aer})) - S_{model} \exp(-(\tau_g + \tau_{aer}))$$

where  $\tau$  is the optical depth due to lines of the gases (subscript *g*), water vapour continuum (*cont*) and aerosols (*aer*), and *S*actual and *S*model are the actual ESS and the ESS used in the model respectively.

Since  $S_{actual}$  is not observed, the authors (in Part II of the paper) perform a Langley analysis on their observations to derive  $S_{Langley}$ , and then apply a calibration constant (*c*) to force  $S_{Langley}$  to agree with  $S_{model}$  (i.e.  $S_{model}=cS_{Langley}$ ). The authors note in Part II that their "closure validation does not provide information on the accuracy of the used ESS" but here we are concerned about the impact of this on the radiance residual.

 $\tau_{cont}$  is then derived from the above equation as

$$\tau_{cont} = -\ln\left(\frac{S_{model}}{S_{actual}}\left(\frac{\Delta I}{S_{model}\exp(-(\tau_g + \tau_{aer}))} + 1\right)\right).$$

If  $S_{model} = S_{actual}$  (i.e. if  $S_{model}$  is indeed the true value), then this equation reduces to the authors' Equation (3). However, if this is not the case, then any error in the ESS (which would lead to a radiance residual even if  $\tau_{cont}$  is zero) gets incorrectly attributed to  $\tau_{cont}$  - the resulting error in  $\tau_{cont}$  is particularly severe for the low values of optical depth found in the window regions, and even the sign of  $\tau_{cont}$  is not constrained to be positive.

We believe that it is important to incorporate the effect of errors/uncertainties in the assumed ESS. We expect that such an analysis will lead to the conclusion that the derived values of the continuum in the centres of the windows are too unreliable to be presented.

2. The consistency between the residual method of deriving the optical depth could be compared with the slopes of the Langley plots in part II, as these are quasi-independent derivations of optical depth (and in particular, the Langley method does not require knowledge of  $S_{actual}$ ).

3. We feel that the summary in the final two sentences of the abstract gives a somewhat misleading picture of the degree of agreement between the new observations and available laboratory measurements. For example, in Figure 1, it is difficult to see that the new measurements are in better agreement with the Bicknell measurements than the FTIR measurements of Ptashnik et al. (2012, 2013). From 5900 to 6600 cm-1, the values derived in this paper, and listed in the Supplement, are almost universally negative, and therefore unphysical. In the 4700 cm-1 region, at the wavenumber of Bicknell's measurements (about 4670 cm-1), the author's central estimate appears as close to the Ptashnik estimate as to Bicknell. Even the comparison with the Mondelain et al. (2015) data is inconclusive. At wavenumbers just below 4250 cm-1, where the authors' data have relatively small error bars, the data points tend to go in between the Mondelain et al. and Ptashnik et al. data. It is only at wavenumbers above 4250 cm-1 that the new data appear to fit better with Mondelain et al., but at these wavenumbers the new observations have too high uncertainties to allow firm conclusions; the upper error-bars nearly overlap the Ptashnik et al. data.

We feel that there would be greater clarity in the abstract if the situation near the band centre is separated from the situation in the window. In the band centres the disagreements between recent FTIR measurements (see especially Paynter et al. (2009)) and MT\_CKD are known to be relatively small, compared to the situation in the windows; these near-band-centre regions

constitute much of the "75%" that is referred to at 1(13). In the windows (e.g. 2800-3000  $\text{cm}^{-1}$  and 4200-4500  $\text{cm}^{-1}$ ), it seems hard to sustain an argument that the new measurements are in any better agreement with MT\_CKD than they are with the Ptashnik FTIR measurements.

**Further comments (co-ordinate system "page number (line number)")**

1(18) and 8(3) We would say "typically a factor of 2-3 times higher". "5" seems an exaggeration to us.

6(18) It would be useful to more clearly state how the solar absorption lines were defined. We assume these were based on the Kurucz ESS described in Part II. However, as noted by Menang et al. (2013) (using both an analysis of their own ground-based observations and using the ACE space-based measurements of Hase et al. (2010)), the Kurucz ESS does not include a number of solar lines that were detected in these two recent works.

7(1-8) We feel it would be useful to produce a plot that showed  $k_{cont}$  using both the linear+constant and the purely constant scaling. At present, the paper has only one figure, and so this could easily be accommodated.

7(29-37) We are unclear why two different temperature dependencies are employed, depending on which laboratory data is used, and what impact it has. Also we were unsure why the MT-CKD temperature dependence was considered more appropriate for some sets than others. It would be useful to see the impact of using a common temperature dependence with all data sets, to establish how much effect this has on the results.

7(40) It may be useful to plot the Paynter and Ramaswamy (2014) data as well as the Baranov and Lafferty (2011) observations.

8(7) There is misleading phrase. There were no "narrow line-like features in the continuum" reported e.g. by Ptashnik et al. (2011); those features were 60 cm-1 (FWHM) broad continuum peaks.

8(9-16) We largely agree with the statements here, but we believe it should be added that the assumption that the foreign continuum has no temperature dependence has not been tested at atmospheric temperatures in the laboratory. And it is that foreign continuum which dominates in the wings of water vapour absorption bands (in particular in the 3200-3400 and 4000-4200 cm-1 regions) where the large and more certain disagreement with FTIR-based results of Ptashnik et al. (2012) is noted by the authors.

8(31) We think that it should be pointed out that there are regions with rather good agreement with Ptashnik et al. (and better than with MT\_CKD), particularly around 3000 cm-1.

13(1) It is rather hard to see the uncertainty bars, especially where they overlap with other data. Perhaps these could be drawn in a bolder format? In addition, we suggest that an additional plot is needed to make clear to the reader that many of the derived values are negative/unphysical; this, of course, cannot be done in a plot with a logarithmic axis.

**References**

- Baranov Yu. I., Lafferty W. J. 2011 The water-vapor continuum and selective absorption in the 3–5 μm spectral region at temperatures from 311 to 363 K. J. Quant. Spectrosc. *Radiat. Transfer.* 112, 1304–1313. doi:10.1016/j.jqsrt.2011.01.024
- Hase, F., L. Wallace, S. McLeod, J. Harrison, and P. Bernath (2010), The ACE-FTS atlas of the infrared solar spectrum. J. Quant. Spectrosc. Radiat. Transfer, 111, 521–528, doi:10.1016/j.jqrst.2009.10.020.
- Menang, K. P., M. D. Coleman, T. D. Gardiner, I. V. Ptashnik, and K. P. Shine (2013), A high-resolution near-infrared extraterrestrial solar spectrum derived from groundbased Fourier transform spectrometer measurements. J. Geophys. Res. Atmos., 118, 5319–5331, doi:10.1002/jgrd.50425.
- Mondelain, D., Vasilchenko, S., Cermak, P., Kassi, S., and Campargue, A.: The self- and foreign-absorption continua of water vapor by cavity ring-down spectroscopy near 2.35 µm. Phys. Chem. Chem. Phys., 17, 27, 17762-1777, doi:10.1039/C5CP01238D, 2015
- Paynter DJ, Ptashnik IV, Shine KP, Smith KM, McPheat R, Williams RG. (2009) Laboratory measurements of the water vapour continuum in the 1200-8000 cm-1 region between 293 K and 351 K. J. Geophys. Res., 114, D21301. doi:10.1029/2008JD011355.
- Thuillier, G., Harder, J. W., Shapiro, A., Woods, T. N., Perrin, J. M., Snow, M., Sukhodolov, T. & Schmutz, W. (2015). The Infrared Solar Spectrum Measured by the SOLSPEC Spectrometer Onboard the International Space Station. *Solar Physics*, **290** (6), 1581-1600 DOI: 10.1007/s11207-015-0704-1.
- Weber, M. (2015). Comment on the Article by Thuillier et al. "The Infrared Solar Spectrum Measured by the SOLSPEC Spectrometer onboard the International Space Station". *Solar Physics*, **290** (6), 1601-1605 DOI 10.1007/s11207-015-0707-y.

---

## Author Response (AR1)

**Author Response, Andreas Reichert, Karlsruhe Institute of Technology, Garmisch-Partenkirchen, Germany, 30 August 2016**

Dear Dr. Maring,

a point-by-point response to the reviews to our manuscript acp-2016-323 is given below and available online at http://www.atmos-chem-phys-discuss.net/acp-2016-323/acp-2016-323-AC1-supplement.pdf. We thank for the sound and very constructive referee and short comments that greatly helped to improve our manuscript. A marked-up version of the manuscript highlighting all changes made by the authors is also attached to this response.

We are confident that all referee comments have been addressed thoroughly and hope that a final publication in ACP is possible soon.

Sincerely,

Andreas Reichert, 30 August 2016

**Author response to the referee and short comments:**

We thank the referees and K. P. Shine et al. for their very sound, constructive and helpful comments which helped us to significantly improve our manuscript. In the following, we provide point-to-point replies to all comments made by the referees. All page and line numbers quoted in this reply refer to the initial version of the manuscript.

**Anonymous Referee #1**

**Comments :**

In Supplementary material the authors give mean water vapor continuum absorption coefficients with negative values which have no physical meaning. Moreover data are given with large uncertainty in the center of the windows so that they are in agreement with all the literature data within the error bars bringing no additional information for those spectral regions. They also weaken the other data obtained in the bands and at the edges of the windows with lower uncertainty. To me it will have more sense to remove large uncertainty data in the center of the 2.1 μm, 1.6 μm windows before publication.
As mentioned in the companion paper Part 1: Very dry atmospheric conditions are a pre-requisite for closure studies of this kind due to the otherwise saturated spectral regions. This has for consequences that in these conditions of dry air, the foreign-continuum represents most of the continuum absorption (more than 70% if the MT_CKD self to foreign ratio is assumed according to the authors).
This can be viewed as a kind of "limitation" of the method but it is also important as there is a real lack of observational constraints for the foreign-continuum.
To conclude, this paper presents state-of the art atmospheric measurements of the continuum which bring interesting information in the near infrared bands and at the edges of the windows mostly for the foreign-continuum for which experimental constraint are clearly

missing. For these reasons this paper deserves to be published in ACP after large uncertainty data in the center of the 2.1 µm and 1.6 µm windows are removed.

*We agree with the referee that in many window regions, our results have too large uncertainties to be useful for comparison to other studies or follow-up analysis. As suggested, we therefore removed data points for which the estimated errors exceed the measured continuum absorption from Fig. 1.*
*However, we respectfully disagree on the statement that the removed measurements have no physical meaning. The mean absorption coefficients have negative values in many window regions, which indicate systematic errors e.g. in AOD measurements or radiometric calibration. However, these possible errors are included in the uncertainty budget and within the given uncertainties, our results are consistent with a positive continuum and with recent continuum quantification studies in these spectral regions. We think that the results may still be valuable since they provide an upper limit to the continuum and therefore included the full results in the revised manuscript the form of an appendix.*

**Specific comments**

**P1, L32**: Burch (1982) and Burch and Alt (1984) used a grating spectrometer for their experiments and not a FTIR spectrometer.

*We thank for pointing out his wording mistake. The manuscript was corrected (Page 1, line 38): "Several studies made use of cell measurements with grating spectrometers (e.g. Burch 1982; 1985; Burch and Alt 1984) or FTIR (Fourier Transform Infrared) spectrometers..."*

The following references are missing for continuum measured:
By CRDS:
- Mondelain D. et al., J. Quant. Spectrosc. Radiat. Transfer, 130, 381 (2013).
- Cormier, J. G. et al., J. Chem. Phys., 122, 114309, (2005)
- Cormier J. G., et al., J. Chem. Phys. 116,1030 (2002)
By OF-CEAS:
- Ventrillard et al. J. Chem. Phys. 143, 134304 (2015)
By calorimetric-interferometry :
- Fulghum, S. F., and M. M. Tilleman, J. Opt. Soc. Am. B Opt. Phys., 8, 2401(1991)

*The additional references were added to the manuscript as suggested by the referee.*

**P1, L38:** The sentence is too general. Cavity enhanced techniques like CRDS and OF-CEAS as well as CI are able to measure continuum absorption in the windows at room temperature (see references above) and even at lower temperature (see Cormier 2005).

*The wording in the manuscript was changed to avoid misrepresentation (Page 1, line 38):*
*"A further challenge for laboratory studies is that they are typically carried out at higher temperatures than those encountered in the atmosphere in order to detect the weak continuum absorption in the limited optical path length of the cells. Note that CDRS and related techniques in principle enable measurements at atmospheric temperature (see e.g. Cormier et al., 2005) but such measurements are not yet available for many spectral regions."*

**P2, L1-3:** The sentences "To date…non-straightforward" are too definitive. Temperature dependences of the self-continuum cross-section have been investigated in different spectral windows (see for example Cormier 2005, Mondelain 2014, Ptashnik 2011, Ventrillard 2015…). In the 2.1 µm window, for example, the temperature dependence measured at high

temperatures by the CAVIAR consortium is similar to that measured at lower temperature by Ventrillard et al. in different part of the window. For sure there is a real lack of observational constraints for the foreign continuum.

*The manuscript was changed as follows to avoid the misleading statement in the initial manuscript (Page 2, line 1):" To date, the temperature dependence of the self-continuum has been investigated by measurements in a number of spectral regions (e.g. Cormier et al., 2005; Mondelain et al., 2014; Ptashnik et al., 2011; Ventrillard et al., 2015). However, the remaining uncertainty of the self-continuum temperature dependence (see e.g. Paynter and Ramaswamy, 2011) and the lack of measurements of the foreign continuum temperature dependence cause considerable uncertainties in the application of the laboratory results on atmospheric radiative transfer calculations."*

**P2, L4-5:** To me it is much more difficult to characterize the continuum from atmospheric spectra than from laboratory spectra recorded in well-known conditions of temperature with sufficiently sensitive and stable techniques. In atmospheric conditions additional uncertainties occur due to water vapor profile, temperature profile, aerosols uncertainties and it is more difficult to separate self and foreign continuum.

*The following discussion was added to the manuscript (Page 4, line 8):" While such atmospheric closure studies enable to avoid some limitations of laboratory measurements as outlined above, they are also subject to a number of major challenges: absorption in the NIR due to aerosols can become comparable to the magnitude of the water vapor continuum absorption of interest (Ptashnik et al., 2015) and requires an accurate separation of continuum and aerosol contribution. Furthermore, the characterization of the atmospheric state (e.g. IWV, water vapor profile, temperature profile, and further trace gas column amounts) is more challenging and typically less accurate than the characterization of experimental conditions in a laboratory study."*

**P2, L23:** Which version of the HITRAN database is used for the line-by-line calculations? Is the line profile used is a Voigt profile truncated at +/- 25 cm-1 from the line center with the plinth subtracted? If yes the authors have to mention it.

*The following text was added to the manuscript (Page 2, line 23): "Spectral line parameters were set according to the aer_v3.2 line list provided alongside the LBLRTM model."*
*Page 2, line 24: "We adopt the definition of the water vapor continuum given in Turner et al. (2010), i.e. water vapor continuum is defined as all absorption by water vapor exceeding a Voight line shape within ± 25 cm-1 of each line center minus the value of the Voight line shape at ± 25 cm-1 ("plinth")."*

**P7, L32**: How exactly calculations are done in the case of Bicknell et al. (2006) as these measurements did not allow dissociating the self from foreign continuum.

*The following text was added to the manuscript (Page 7, line 32): "Since the results of Bicknell et al. (2006) do not allow a dissociation of self from foreign continuum is not possible, we assumed the self-to-foreign ration suggested by the MT_CKD model to calculate the corresponding value of $\bar{k}_{cont}$. "*

**P7, L8**: Assuming that the partitioning in self and foreign continuum given by the MT_CKD model is a quite strong hypothesis as the very few laboratory studies of the foreign-continuum (Ptashnik PTRSA 2012 and Mondelain PCCP 2015) seem to show that the foreign continuum cross-sections are largely underestimated by MT_CKD in the windows.

*The following text was added to the manuscript (Page 7, line 8): "Note that this assumption has to be considered tentative since for both self- and foreign continuum the results of recent laboratory studies deviate from the MT_CKD model especially in window regions (e.g. Ptashnik et al, 2012, 2013; Mondelain et al., 2015)."*

**P7, L38:** The very good agreement with CRDS-based measurements of Mondelain et al. (2015) is essentially a good agreement with the foreign-value measured in this paper due to the dominating contribution of the foreign-continuum in the atmospheric conditions encountered at Zugspitze. This is an important point as in the Mondelain et al study the measured foreign-cross section is 4.5 times larger than the one given in MT_CKD 2.5. The fact that in the 4100 to 4200 cm$^{-1}$ spectral range the MT_CKD model underestimate the continuum goes in the same direction as well as the Ptashnik 2012 paper. This point should be underlined by the authors.

*The following discussion was added to the manuscript (Page 7, line 38):"Due to the dominant role of the foreign continuum in the 4100 to 4200 cm$^{-1}$ spectral range, this agreement mainly corresponds to a comparison of the foreign continuum results of Mondelain et al. (2015) and our measurements. Therefore, our results are consistent with the finding of Mondelain et al. (2015) and Ptashnik et al. (2012) that the foreign continuum is underestimated by the MT_CKD model in this spectral region."*

**Fig 1**: Some literature data at room temperature are not plotted on this figure (Ventrillard 2015, Mondelain 2014). It will be good to incorporate them in the figure.

*The intention of Fig. 2 (previously Fig. 1) was to present a comparison of our results to measurements which comprise a quantification of the self and foreign continuum at or below room temperature using the same (as e.g. for Mondelain et al., 2015) or a very similar experimental setup (as for Ptashnik et al. 2012, 2013).*
*To clarify this intention, the following text was added to the manuscript (Page 6, line 30):*
*"The figure includes laboratory measurements carried out at or below room temperature which provided constraints on both the self and foreign continuum using the same or a very similar experimental setup."*
*The measurements of Ventrillard et al. 2015 and Mondelain et al., 2014 were not included in the figure since they only comprise results for the self continuum, while additional assumptions have to be made on the foreign continuum to calculate the overall continuum absorption. However, we agree on the relevance of these studies and have therefore referenced and discussed them in the manuscript.*

**Technical corrections:**

**P2, L25**: closure experiment und the related uncertainties: und->and.

*The manuscript was changed as suggested*

**Referee #2, Penny Rowe**

**Main comments:**

1) Literature review.   I do not think it is necessary to exhaustively reference laboratory work performed over a different spectral range.  But all lab work that overlaps the spectral region should be referenced. (Thus I think some, but not all, of the references suggested by the

other reviewer need to be included). In addition, studies in atmospheric conditions with similar instruments should be referenced (although they are mainly in different spectral regions):

**Page 2, line 5.** Please include the following references with a sentence such as, "The continuum has been investigated for atmospheric conditions using measurements of atmospheric emitted infrared radiance for other spectral regions (Tobin et al., 1999; Rowe and Walden, 2009) and for part of the region of interest for this study (Newman et al. 2011; 2400 to 3200 cm-1), but not for the spectral region 3200 to 7800 cm-1."
- Tobin et al.: Downwelling spectral radiance observations at the SHEBA ice station: Water vapor continuum measurements from 17 to 26 micron, J. Geophys. Res., 104, 2081-2092, 1999.
- Rowe, P.M. and Walden, V.P.: Improved measurements of the foreign-broadened continuum of water vapor in the 6.3 micron band at -30 C, Appl. Opt. 48, 1358-1365, 2009.
- Newman et al. 2012: Airborne and satellite remote sensing of the mid-infrared water vapour continuum, Phil. Trans. R. Soc. A 370, 2611-2636.

*The following literature review was added to the manuscript as suggested by the referee (**page 2, line 5**):*
*"The continuum has been investigated for atmospheric conditions using measurements of atmospheric emitted infrared radiance for other spectral regions (e.g. Tobin et al., 1999; Rowe and Walden, 2009). However, atmospheric measurements are available only for a fraction of the spectral region covered by this study (Newman et al. 2011; 2400 to 3200 cm$^{-1}$), while for the remaining interval from 3200 to 7800 cm$^{-1}$, no atmospheric measurements have been reported."*

The results of Newman et al. should also be discussed (e.g. on Page 7, line 38).

*The following discussion was added to the manuscript (**page 7, line 38**):*
*"A fraction of the spectral range covered by this study, namely 2500 -3200 cm$^{-1}$, was also included in the airborne measurements by Newman et al. (2011). Newman et al. (2011) conclude that the increase of the self continuum in MT_CKD 2.5 compared to MT_CKD 2.4 lead to reduced spectral residuals, while no firm conclusion can be drawn in the 2500 - 3200 cm$^{-1}$ –range on whether MT_CKD 2.5 or the results of Ptashnik et al. (2011) represent a more appropriate quantitative description of the water vapor self continuum. These findings are in agreement to the results of this study, given that both are not consistent with continuum absorption being weaker than indicated by MT_CKD 2.5. "*

Also on **Page 8, line 20**, change "in this spectral range" to "for most of this spectral range."

*The manuscript was changed as suggested.*

2) Fig. 1. If you also include kcont for the self and foreign-broadened parts of the MT-CKD continuum separately in this plot, it will show the relative importance of each for your results. If any of the lower bounds on your error bars are significantly different from zero, you might make those error bars black instead of gray so they stand out more. In the caption, "(black)" needs to be changed to "(black; gray error bars are shown for points for which only the upper threshold can be determined to within the uncertainty)." If you have measurements that are not shown on this plot, you should create a second panel below on a linear y-scale where they can be seen. In the caption, state something like: "x points that fall outside the plot region are not shown in the upper panel (log scale) but are evident in the lower panel (linear scale)"

*The contributions from self- and foreign broadened MT_CKD-continuum were included in Fig. 1 as suggested. Only data points with significant continuum absorption were included in Fig. 1 (as suggested by referee #1), while the entire set of results is shown in the supplementary figure Fig. B1*

3) (Optional) The paper would likely receive more citations if the following changes were made.

- Title:  Stating the spectral range explicitly in the title will help readers more quickly determine if the paper is of interest to them, especially given the large spectral range. (You could omit "under atmospheric conditions" because the title already includes "radiative closure experiment.")

*The title was changed to: "Quantification of the mid- and near-infrared water vapor continuum in the 2500 to 7800 $cm^{-1}$ spectral range under atmospheric conditions"*

- Cf.  If you estimate the foreign-broadened part of the continuum (Cf), it can be compared to previous work, and incorporated (e.g. in figures) in future publications by other authors. I suggest estimating the self-broadened continuum (Cs) based on what you think is most accurate (MT-CKD or other; give rationale) and the atmospheric water and temperature structure, removing its effects from $k_{cont}$, and calculating Cf. Increase your error bars correspondingly (uncertainty estimate for self-broadened continuum x 0.1 to 0.3). Discuss how you calculated Cf briefly in the text. Add a figure showing the subset of results for which the error bars are small enough to be useful.

*The following text was added to the manuscript (Page 7, line 12):*
*"If values for $c_f$ are required, further assumptions on the self continuum have to be made before subtracting this contribution. As an example, Supplement B to this manuscript contains a list of $c_f$-values for all spectral bins where $c_f$ exceeds the uncertainty estimate. The results were calculated from our measurements assuming the self continuum to be consistent with the MT_CKD model. Recent laboratory measurements (e.g. Ptashnik et al., 2013) suggest that this assumption may not be appropriate. However, alternative sources of the self continuum neither constitute a more robust estimate, given the inconsistencies between different laboratory results, the uncertainty of the self continuum temperature dependence and the fact that the foreign continuum is likely to be the dominant contribution to the overall continuum absorption for the dry atmospheric conditions of our study and the spectral windows covered by the measurements (see Fig. 1).  A 50 % uncertainty was assumed for the self continuum as suggested by Paynter et al. (2011) and is included in the uncertainty of $c_f$ in addition to the uncertainty budget presented in Sect. 3.2."*

**Minor comments**

- Give the wavenumber range the first time you mention each region (near-infrared, etc)

*The wavenumber ranges were added to the manuscript. **Page 1, line 23**: NIR, "4000-14000 $cm^{-1}$", **page 2, line 11**: "FIR, 2-667 $cm^{-1}$", **ib**. "MIR, 667-4000 $cm^{-1}$"*

- **Page 4, Line 8**: Please show examples of measured and synthetic radiance spectra. (You can alternatively reference your other paper here, but I think it would be nice to have it here as well).

*The following text was added to the manuscript (**page 4, line 9**):*
*"Figure 1 shows the mean measured and synthetic radiance spectra for the closure data set that will be presented in Sect. 3.3."*

*A figure showing the mean measured and synthetic radiance spectra was added to the manuscript (Fig. 1):*

[Figure]

**Fig. 1**: Mean measured (black) and synthetic (red) radiance spectra for the closure data set selected according to the criteria presented in Sect. 3.3.

- **Page 4, line 20 (approx.)**. You might mention here that cs is strongly temperature dependent but that cf is thought to be only weakly temperature dependent.

*The following text was added to the manuscript (**page 4, line 26**):*
*"In addition to their different dependence on water vapor density according to Eq. 5, self- and foreign-broadened continua are characterized by their distinct temperature dependence: while the self continuum shows strong negative temperature dependence, the foreign continuum is assumed to have no or only weak temperature dependence."*

**Technical and grammar corrections**

- **Page 1, Line 26**, remove the word "exact"
- **Page 1, Line 31**, change "both continuum" to "continuum absorption, including the contributions of both the self and foreign-broadened continuum"
- **Page 2, Line 10**, change "thereafter" to "hereafter"
- **Page 3, Line 4**, add the word "for" before "data"
- **Page 3, line 8**, change "disposes" to "consists of" or "includes"
- **Page 3, line 9**, rephrase "centered at nm." Perhaps give the range of the channels or the bounds of each.

***Page 3, line 9** was rephrased as follows: "Only information from 5 channels whose central wavelengths are in the spectral region between 439.6 and 781.1 nm was used in the analysis. The exact filter wavelengths and full width at half maximum (FWHM) values of these channels are listed in Table 1."*

- **Page 3, line 24**, Do you mean errors in the AOD measurements from the sun photometer (rather than errors in the sun photometer measurements)? If so, add "AOD

determined from" before "sun photometer measurements."

"AOD determined from" was added on **page 3, line 24**

- **Page 3, lines 26-27**, "The following measurements." The sentence is awkward, rephrase.

*The sentence was rephrased as follows **(page 3, line 24): "The AOD uncertainty comprises several contributions: First of all, the AOD determined from the sun photometer measurements…"***

- **Page 4, line 11**, change "the criteria presented" to "criteria that will be presented"
- **Page 6, line 6**, change "requested" to "required"
- **Page 7, line 21**, change "where treated in sufficiently" to "were treated in a sufficiently"
- **Page 8, line 18**, change "presented" to "present"
- **Page 9, line 12**, change "we thank for support by the" to "we are grateful for support by the"
- **Table A1**. Convert into two tables, putting further parameters in a separate table.

*All technical and grammar corrections were applied to the manuscript as suggested by the referee.*

**Short comment by Shine et al.**

Reichert and Sussmann (2016) present an important attempt to characterise the water vapour continuum in the near-infrared in atmospheric conditions. Given that relatively few such measurements exist, such work is very welcome.

We have a number of comments on the paper. The major one relates to our comment on Part II of this paper, where the authors calibrate their measurements to an assumed extraterrestrial solar spectrum (ESS); as we note in that comment, there are significant uncertainties in the ESS. This uncertainty has important consequences for the derivation of the continuum, especially in the window regions, which are not taken into account here.
It is our view that this uncertainty renders the continuum derivations here unreliable in window regions; the fact that many of the derived continuum values in the windows are negative and therefore unphysical (as shown in the data in their Supplement but not in the figure in the paper) adds support to the opinion given by Reviewer 1 (10.5194/acp-2016-323-RC1) that the derived continuum values deep in the window are so uncertain that they should not be presented.

*Major comments*

*1. Equations (2) and (3) derive the continuum optical depth from the difference between the observed downward radiance at the surface and the modelled radiance ignoring the continuum. To do this reliably requires that the ESS is well constrained. This is not currently the case, as we explain in our comment in Part II (see e.g. Thuillier et al. (2015) and Weber (2015)).*
*Various derivations from satellite and other observations differ by 5 - 10%. The authors' method is essentially to write a radiance residual (their Equation (2)) between observations and model so that*

$$\Delta I = S_{actual} \exp(-(\tau_g + \tau_{cont} + \tau_{aer})) - S_{model} \exp(-(\tau_g + \tau_{aer}))$$

*where τ is the optical depth due to lines of the gases (subscript g), water vapour continuum (cont) and aerosols (aer), and $S_{actual}$ and $S_{model}$ are the actual ESS and the ESS used in the model respectively.*

*Since $S_{actual}$ is not observe, the authors (in Part II of the paper) perform a Langley analysis on their observations to derive $S_{Langley}$, and then apply a calibration constant (c) to force $S_{Langley}$ to agree with $S_{model}$ (i.e. $S_{model} = cS_{Langley}$). The authors note in Part II that their "closure validation does not provide information on the accuracy of the used ESS" but here we are concerned about the impact of this on the radiance residual.*

*$τ_{cont}$ is then derived from the above equation as*

$$\tau_{cont} = -\ln\left(\frac{S_{model}}{S_{actual}}\left(\frac{\Delta I}{S_{model}\exp\left(-(\tau_g+\tau_{aer})\right)}+1\right)\right).$$

*If $S_{model} = S_{actual}$ (i.e. if $S_{model}$ is indeed the true value), then this equation reduces to the authors' Equation (3). However, if this is not the case, then any error in the ESS (which would lead to a radiance residual even if $τ_{cont}$ is zero) gets incorrectly attributed to $τ_{cont}$ – the resulting error in $τ_{cont}$ is particularly severe for the low values of optical depth found in the window regions, and even the sign of $τ_{cont}$ is not constrained to be positive.*

*We believe that it is important to incorporate the effect of errors/uncertainties in the assumed ESS. We expect that such an analysis will lead to the conclusion that the derived values of the continuum in the centres of the windows are too unreliable to be presented.*

*We thank K. P. Shine et al. for pointing out recent research that indicates that the ESS uncertainty may be significantly higher than assumed in our initial manuscript. An extensive discussion of recent ESS results has been included the companion publication Part II which introduces the radiometric calibration method used in our analysis.*

*In their comment, Shine et al. raise the question whether the continuum results might be significantly influenced by inaccuracies in the used ESS. A discussion of this important point was missing in the initial version of the manuscript, and we thank Shine et al. for highlighting this issue.*

*In their first equation, Shine et al. make a definition of the spectral residual ΔI. Based on this definition, they conclude that errors in the ESS might have significant influence on the determined continuum OD.*

*We agree that the analysis made by Shine et al. is correct and their first equation represents a valid definition of the spectral residual. However, the definition of spectral residuals made in Eq. 2 of our manuscript, which corresponds to the continuum quantification procedure in our study, is not fully equivalent to the definition given by Shine et al.*

*While the first term of the definition of Shine et al. contains the 'real' downwelling solar radiance (i.e. without ESS errors), our definition relies on $I_{FTIR}$, i.e. the measured downwelling radiance. The measured radiance according to Eq. 2 of our manuscript already contains ESS errors due to the use of the ESS in the radiative calibration. Note that both the modeled and the measured radiance rely on the same ESS, which as noted rightly by Shine et al. may be prone to substantial inaccuracies. The design of the continuum quantification procedure used in our study thereby greatly reduced the impact of ESS errors on the continuum results.*

*The influence of ESS errors on the continuum results for the continuum quantification method presented in our manuscript can be described as follows:*

*a) For spectral points where only Langley data was used for the calibration, ESS errors have no influence on the continuum results. The situation at these points is equivalent to the well-known Langley calibration of sun photometers where no information on the solar spectrum is needed to infer accurate atmospheric optical depth.*

*b) In between the Langley points, where additional blackbody measurements are used for calibration, there is in fact a second-order influence of ESS errors on our continuum results. To investigate the magnitude of this effect, we repeated the continuum quantification analysis presented in the manuscript using the high-resolution ESS by Menang et al. (2013) instead*

*of the ESS by Kurucz (2005). Apart from the exchange of the ESS, the data set and analysis was not modified. This is a good test to assess the sensitivity of the results to ESS uncertainty since the ESS by Menang et al. (2013) differs from the Kurucz-ESS by about 5 % (recent ESS results generally feature differences of up to ±5 % compared to the Kurucz-ESS) and features many solar lines not included in the Kurucz-ESS. Note that the Menang-ESS covers the spectral range v > 4000 cm$^{-1}$. The comparison analysis is therefore limited to wavenumbers greater than the first Langley calibration point covered by the Menang-ESS, namely v > 4200 cm$^{-1}$.*

*The exchange of the ESS has only minor influence on the continuum results as visible in Fig. 3 and Fig. 4 (which uses a linear scale to show the situation in the window regions). On average, the continuum absorption coefficient determined using the Menang-ESS (blue data points) differs from the results obtained with the Kurucz-ESS (black data points) by 11.1 % of the continuum uncertainty estimate. We use this difference as an estimate of the ESS-related continuum uncertainty and include it as an additional contribution in the continuum uncertainty budget presented in the companion paper Part I.*

*The following text was added to the manuscript to discuss the ESS-induced uncertainty of the continuum results (Page 7, line 12):*

*"As outlined in the companion paper Part II, recent studies on the NIR ESS have yielded results that feature differences of up to 5-10% (see e.g. Menang et al, 2013; Bolsee et al, 2014; Thuillier et al., 2014, 2015; Weber et al. 2015). Furthermore, the number of solar lines differs significantly e.g. between the ESS versions of Kurucz (2005) and Menang et al., (2013). To investigate the influence of inaccuracies in the ESS on the continuum results, the continuum retrieval was repeated using the ESS determined by Menang et al. (2013) instead of the ESS by Kurucz (2005) that was used to generate the results presented in Fig. 2. This is a good test to assess the sensitivity of the results to ESS uncertainty since these ESS versions differ by about 5 %, while recent ESS results generally feature differences of up to ±5 % compared to the ESS of Kurucz (2005). Note that the Menang et al. (2013) ESS only covers the spectral region > 4000 cm$^{-1}$. The comparison is therefore restricted to 4233 cm$^{-1}$ < v < 7800 cm$^{-1}$, which corresponds to the first Langley point covered by the Menang et al. (2013) ESS and the maximum wavenumber value covered by our analysis. For this region, the median of the absolute value of the difference between the Menang et al. (2013) and Kurucz (2005) continuum results corresponds to 11.1% of the continuum uncertainty estimate. Therefore, ESS uncertainty does not constitute a major accuracy limitation of our analysis, which his due to the fact that the same ESS is used for both synthetic spectra calculation and the radiometric calibration presented in the companion paper Part II. The ESS-related continuum uncertainty was estimated from the difference of the Menang et al. (2013) and Kurucz (2005) results and included in the uncertainty budget as described in Sect. 3.2 (see also Fig. 9 of the companion paper Part I). For the spectral region v < 4233 cm$^{-1}$, where no direct comparison is available, the ESS-induced continuum uncertainty was assumed to correspond to 11.1% of the remaining overall uncertainty as suggested by the median value in the spectral range v > 4233 cm$^{-1}$."*

[Figure]

**Fig. 3**: Mean continuum absorption coefficient derived with the method and data set described in Sect. 3 using the ESS by Menang et al. (2013) (blue data points) and by Kurucz (2005) (black data points). The different ESS sources differ by about 5 % and many solar lines not present in Kurucz (2005) were included in Menang et al. (2013).

[Figure]

**Fig. 4**: Same as Fig. 3 but using a linear scale to show the results in the window regions.

2. The consistency between the residual method of deriving the optical depth could be compared with the slopes of the Langley plots in part II, as these are quasi - independent derivations of optical depth (and in particular, the Langley method does not require knowledge of $S_{actual)}$.

*This consistency check had been performed by the authors but was omitted in the initial manuscript for the sake of brevity. The Langley results on the continuum are consistent with*

*the results derived from calibrated spectra throughout 98.0 % of the 2500 to 7800 cm⁻¹ spectral range (see figure below) and the corresponding results are included in the revised manuscript as an appendix (Appendix C).*

[Figure]

**Fig. C1.:** Mean continuum absorption coefficient $\overline{k}_{cont}$ determined from 12 December 2013 spectra using the Langley method and corresponding 2 σ uncertainties (black). Results are compared to the calibrated method for the same spectral dataset (orange) and the MT_CKD 2.5.2 model (blue).

3. We feel that the summary in the final two sentences of the abstract gives a somewhat misleading picture of the degree of agreement between the new observations and available laboratory measurements. For example, in Figure 1, it is difficult to see that the new measurements are in better agreement with the Bicknell measurements than the FTIR measurements of Ptashnik et al. (2012, 2013). From 5900 to 6600 cm⁻¹, the values derived in this paper, and listed in the Supplement, are almost universally negative, and therefore unphysical. In the 4700 cm⁻¹ region, at the wavenumber of Bicknell's measurements (about 4670 cm⁻¹), the author's central estimate appears as close to the Ptashnik estimate as to Bicknell.

Even the comparison with the Mondelain et al. (2015) data is inconclusive. At wavenumbers just below 4250 cm⁻¹, where the authors' data have relatively small error bars, the data points tend to go in between the Mondelain et al. and Ptashnik et al. data. It is only at wavenumbers above 4250 cm⁻¹ that the new data appear to fit better with Mondelain et al., but at these wavenumbers the new observations have too high uncertainties to allow firm conclusions; the upper errorbars nearly overlap the Ptashnik et al. data.

We feel that there would be greater clarity in the abstract if the situation near the band centre is separated from the situation in the window. In the band centres the disagreements between recent FTIR measurements (see especially Paynter et al. (2009)) and MT_CKD are known to be relatively small, compared to the situation in the windows; these near – band - centre regions constitute much of the "75%" that is referred to at 1(13). In the windows (e.g. 2800 – 3000 cm⁻¹ and 4200 - 4500 cm⁻¹), it seems hard to sustain an argument that the new measurements are in any better agreement with MT_CKD than they are with the Ptashnik FTIR measurements.

*The abstract was changed as follows to avoid a misleading impression on the degree of agreement (Page 1, line 15): "In the wings of water vapor absorption bands, our measurements indicate about 2-5 times stronger continuum absorption than MT_CKD, namely in the 2800 to 3000 cm⁻¹ and 4100 to 4200 cm⁻¹ spectral ranges. The measurements*

*are consistent with the laboratory measurements of Mondelain et al. (2015), which rely on cavity ring-down spectroscopy (CDRS), and the calorimetric-interferometric measurements of Bicknell et al. (2006). Compared to the recent FTIR laboratory studies of Ptashnik et al. (2012, 2013), our measurements are consistent within the estimated errors throughout most of the spectral range. However, in the wings of water vapor absorption bands our measurements indicate typically 2 – 3 times weaker continuum absorption under atmospheric conditions, namely in the 3200 to 3400 cm$^{-1}$, 4050 to 4200 cm$^{-1}$, and 6950 to 7050 cm$^{-1}$ spectral regions."*

**Further comments**
(coordinate system "page number(line number)")

**1(18) and 8(3)** We would say "typically a factor of 2 - 3 times higher". "5" seems an exaggeration to us.

*Since differences by a factor of 5 between our results and the findings of Ptashnik et al. (2012, 2013) only occur at few spectral points, the manuscript was changed as suggested.*

**6(18)** It would be useful to more clearly state how the solar absorption lines were defined. We assume these were based on the Kurucz ESS described in Part II. However, as noted by Menang et al. (2013) (using both an analysis of their own ground - based observations and using the ACE space - based measurements of Hase et al. (2010)), the Kurucz ESS does not include a number of solar lines that were detected in these two recent works.

*The following more extensive discussion of solar line removal was included in the revised manuscript (Page 6, line 18):" ii) Regions around solar lines were excluded. This was implemented as an exclusion of all points for which the extra-atmospheric solar radiance according to the ESS of Kurucz (2005) is more than 0.5 % below the upper envelope. Note that recent studies indicate that many solar lines are missing in this ESS (see Menang et al., 2013). However, solar lines omitted in the ESS of Kurucz (2005) are discarded from further analysis by applying the selection criterion (i). As outlined in Sect. 4, a repetition of the continuum analysis using the ESS of Menang et al. (2013), which includes many additional solar lines only leads to very minor changes in the continuum results, thereby indicating that the solar line removal scheme according to criteria (i) and (ii) is appropriate."*

**7(1-8)** We feel it would be useful to produce a plot that showed $k_{cont}$ using both the linear+constant and the purely constant scaling. At present, the paper has only one figure, and so this could easily be accommodated.

*The analysis on page 7, line 1-8 covers the scaling of $k_{cont}$ with respect to IWV, which furthermore depends on wavenumber. Therefore, a complete representation of the data either requires a three-dimensional plot with axes v, IWV, and $k_{cont}$ or two-dimensional plots with axes IWV and $k_{cont}$ for each of the 532 wavenumber bins. The first alternative is not suitable for a figure due to the large number of data points in one plot, while the second alternative requires excessive space. We therefore decided to include supplementary figures in the revised manuscript (Fig. A3) showing the measured $k_{cont}$ and the best fit linear and constant scaling for three representative wavenumber bins: 7200 cm$^{-1}$ (within water vapor absorption band), 4100 cm$^{-1}$ (wing of band), and 2700 cm$^{-1}$ (window region).*

**7(29-37)** We are unclear why two different temperature dependencies are employed, depending on which laboratory data is used, and what impact it has. Also we were unsure

why the MT-CKD temperature dependence was considered more appropriate for some sets than others. It would be useful to see the impact of using a common temperature dependence with all data sets, to establish how much effect this has on the results.

*In the updated version of Fig. 2, the temperature dependence of Rädel et al. (2015) is used for all laboratory measurements to improve consistency.*

**7(40)** It may be useful to plot the Paynter and Ramaswamy (2014) data as well as the Baranov and Lafferty (2011) observations.

*The data of Paynter and Ramaswamy (2014) was added to Fig 2. The following discussion of the results of Paynter and Ramaswamy (2014) was added to the manuscript (Page, line):*
*"The comparison of our results to the BPS_MTCKD 2.0 continuum proposed by Paynter and Ramaswamy (2014) is mostly equivalent to the comparison to MT_CKD. This is due to the fact that the BPS_MTCD 2.0 foreign continuum, which constitutes the dominant contributor for the dry atmospheric conditions encountered in our data set, was mostly adopted from MT_CKD. Exceptions include the spectral regions from 2500 to 3000 $cm^{-1}$, 5200 to 5600 $cm^{-1}$, and 6800 to 7000 $cm^{-1}$, where our results show better consistency with the MT_CKD 2.5.2 model"*
*The intention of Fig. 2 (previously Fig. 1) was to present a comparison of our results to measurements which comprise a quantification of the self and foreign continuum at or below room temperature using the same or a very similar experimental setup. To clarify this intention, the following text was added to the manuscript (Page 6, line 30):*
*"The figure includes laboratory measurements carried out at or below room temperature which provided constraints on both the self and foreign continuum using the same or a very similar experimental setup."*
*We therefore did not include the results of Baranov (2011) and Baranov and Lafferty (2011) in Fig. 2 which were obtained at > 311 K for the self continuum and > 326 K for the foreign continuum. However, the following discussion of the results of Baranov (2011) and Baranov and Lafferty (2011) was added to the revised manuscript (Page 8, line 4): Further FTIR laboratory measurements were carried out by Baranov and Lafferty (2011) at temperatures of 311 - 363 K on the self continuum and by Baranov (2011) at 326 – 363 K on the foreign continuum at v < 3500 $cm^{-1}$. The results of these studies generally agree well within the estimated errors with the findings of Ptashnik et al. (2012, 2013).*

**8(7)** There is misleading phrase. There were no "narrow line – like features in the continuum" reported e.g. by Ptashnik et al. (2011); those features were 60 $cm^{-1}$ (FWHM) broad continuum peaks.

*The misleading text was removed from the manuscript.*

**8(9-16)** We largely agree with the statements here, but we believe it should be added that the assumption that the foreign continuum has no temperature dependence has not been tested at atmospheric temperatures in the laboratory. And it is that foreign continuum which dominates in the wings of water vapour absorption bands (in particular in the 3200 – 3400 and 4000 – 4200 cm-1 regions) where the large and more certain disagreement with FTIR - based results of Ptashnik et al. (2012) is noted by the authors.

*The following text was added to the manuscript (Page 8, line 16):*
*"Note, however that the assumption that the foreign continuum has no significant temperature dependence, which was used in the data analysis, has not been robustly confirmed by measurements under atmospheric conditions yet. Due to the dominant role of the foreign continuum in the wings of water vapor absorption bands, inaccuracies in the foreign continuum temperature dependence would have a significant influence on the conversion of the findings of Ptashnik et al. (2012, 2013) to atmospheric temperatures."*

**8(31)** We think that it should be pointed out that there are regions with rather good agreement with Ptashnik et al. (and better than with MT_CKD), particularly around 3000 cm$^{-1}$

*The following text was added to the manuscript (Page 8, line 31): "There are also several regions where our results are in good agreement with the findings of Ptashnik et al. (2012, 2013), most notably around 3000 cm$^{-1}$."*

**13(1)** It is rather hard to see the uncertainty bars, especially where they overlap with other data. Perhaps these could be drawn in a bolder format?
In addition, we suggest that an additional plot is needed to make clear to the reader that many of the derived values are negative/unphysical; this, of course, cannot be done in a plot with a logarithmic axis.

*A darker color and bolder format was used for the error bars in the revised manuscript. The additional Fig. 4 now shows a non-logarithmic representation of the continuum results.*

[revised manuscript text omitted]